# Long-term studies of the summer wind in the mesosphere and lower thermosphere at middle and high latitudes over Europe

Juliana Jaen[1], Toralf Renkwitz[1], Huixin Liu[2], Christoph Jacobi[3], Robin Wing[1], Aleš Kuchař[4], Masaki Tsutsumi[5], Njål Gulbrandsen[6], and Jorge L. Chau[1]

[1]Leibniz-Institute of Atmospheric Physics at the University of Rostock, Schloss-Strasse 6, 18225 Kühlungsborn, Germany
[2]Department of Earth and Planetary Science, Kyushu University, Fukuoka, Japan
[3]Institute for Meteorology, Leipzig University, Stephanstr. 3, 04103 Leipzig, Germany
[4]Institute for Meteorology and Climatology, University of Natural Resources and Life Sciences (BOKU), Vienna, Austria.
[5]National Institute of Polar Research, Tokyo, Japan
[6]Tromsø Geophysical Observatory, UiT - The Arctic University of Norway, Tromsø, Norway

**Correspondence:** Juliana Jaen (jaen@iap-kborn.de), Huixin Liu (liu.huixin.295@m.kyushu-u.ac.jp)

**Abstract.** Continuous wind measurements using partial reflection radars and specular meteor radars have been carried out for nearly two decades (2004-2022) at middle and high latitudes over Germany ($\sim$54°N) and northern Norway ($\sim$69°N), respectively. They provide crucial data for understanding the long-term behavior of winds in the mesosphere and lower thermosphere. Our investigation focuses on the summer season, characterized by the low energy contribution from tides and relatively stable stratospheric conditions. This work presents the long-term behavior, variability and trends of the maximum velocity of the summer eastward, westward and southward winds. In addition, the geomagnetic influence on the summer zonal and meridional wind is explored at middle and high latitudes. The results show a mesospheric westward summer maximum located around 75 km with velocities of 35-54 m/s, while the lower thermospheric eastward wind maximum is observed at $\sim$97 km with wind speeds of 25-40 m/s. A weaker southward wind peak is found around 86 km ranging from 9-16 m/s. The findings indicate significant trends at middle latitudes in the westward summer maxima with increasing winds over the past decades, while the southward winds show a decreasing trend. On the other hand, only the eastward wind in July has a decreasing trend at high latitudes. Evidence of oscillations around 2-3, 4 and 6 years modulate the maximum velocity of the summer winds. Particularly a periodicity between 10.2-11.3 years found in the westward component is more significant at middle latitudes than at high latitudes, possibly due to solar radiation. Furthermore, stronger geomagnetic activity at high latitudes causes an increase in eastward wind velocity, whereas the opposite effect is observed in zonal jets at middle latitudes. The meridional component appears disturbed during high geomagnetic activity, with a notable decrease in the northward wind strength below approximately 80 km at both latitudes.

## 1 Introduction

The Earth's atmosphere constitutes a complex and dynamic system. The mesosphere and lower thermosphere (MLT) spanning between 50 and 110 km is a region where the neutral and ionized atmosphere coexists. The ionization process due to the absorption of solar irradiance governs the thermosphere, whereas the neutral atmosphere is subjected to active winds and wave

interactions, leading to chemical mixing and temperature regulation. The stratosphere, which is located below the mesosphere, is the region where ozone mainly absorbs the ultraviolet (UV) radiation from the Sun.

In the 1980s the ozone hole was discovered, which led to the awareness of health problems due to UV, radiation. In 1987 the Montreal Protocol was implemented to stop the emission of ozone-depleting substances and studies show a positive shift in the trends of stratospheric ozone in 1995 at equatorial latitudes and 2000 at high latitudes (e.g., Weber et al., 2022). Since then, researchers have been studying long-term compositions, temperatures and dynamics to understand the behavior of the atmosphere and the human footprint on the environment. Greenhouse gases, including $CH_4$, $H_2O$, $O_3$ and $CO_2$ are studied as tracers to monitor the evolution of the atmosphere (Bremer and Berger, 2002; Bremer and Peters, 2008; Yue et al., 2015; Peters and Entzian, 2015; Qian et al., 2017; Peters et al., 2017; Karagodin-Doyennel et al., 2021, etc.). At an altitude of 96 km, atomic oxygen is formed through the photo-dissociation of $O_2$ and $O_3$ in the mesopause. The atomic oxygen then interacts with $CO_2$ through collisions, resulting in a radiative cooling effect that leads to hydrostatic contraction of the atmosphere (Gu et al., 2022; Akmaev, 2002; Li et al., 2021; Pisoft et al., 2021; Dawkins et al., 2023, etc.). As a consequence, carbon dioxide serves as a reliable indicator of cooling in the middle atmosphere, even during periods of disturbed geomagnetic activity (e.g., Liu et al., 2020).

The dynamics of the MLT are governed by the interaction of mean winds and waves from large-scale planetary waves to small-scale gravity waves. The latter is driven by gravity and buoyancy in the atmosphere and is generated by orographic forcing, convection, wind shear, or wave interaction (Fritts and Alexander, 2003). Most gravity waves are generated in the troposphere and propagate upward and horizontally, breaking already in the troposphere and lower stratosphere. During winter the zonal-mean zonal wind in the stratosphere and mesosphere is eastward and reverses in summer to westward. The mean wind flow is crucial for the propagation of gravity waves. The linearized theory explains that gravity waves with an eastward velocity phase filtered by the westward wind reach the mesopause, where they break and deposit the momentum that decelerates the mean flow. This deceleration causes a wind reversal from westward to eastward in the lower thermosphere. As a consequence of the injection of energy from the breaking of the gravity waves and under the Coriolis force, a mean meridional circulation is induced from the summer hemisphere to the winter hemisphere, generating an upwelling in summer and a downwelling in winter. This circulation is the cause of the cold (warm) summer (winter) mesopause (Andrews et al., 1987; Holton and Alexander, 2000; Holton, 2004). In a non-linear regime, the contribution of anisotropic gravity waves has been proven to deposit a significant amount of momentum to the mean flow at lower altitudes (Medvedev et al., 1998). Furthermore, regions characterized by intense wind jets exhibit significant anisotropy, particularly in the upper area of a strong wind jet (Warner et al., 2005; Gong et al., 2008), which is a characteristic of the summer MLT. Additionally, the MLT exhibits sensitivity to external phenomena such as the stratospheric quasi-biennial oscillation (QBO), which alters the direction of the zonal winds over a span of 26-28 months, as well as the equatorial ocean-atmospheric warming (and cooling) that occurs during the northern hemisphere winter season. This phenomenon, referred to as El Niño-southern oscillation (ENSO), has periods that are not precisely defined but generally span around three to six years (Baldwin et al., 2001; Jacobi and Kürschner, 2002; Wang and Picaut, 2004; Espy et al., 2011; Offermann et al., 2015; French et al., 2020; Jaen et al., 2022).

Sprenger and Schminder (1969) studied the wind at 95 km during winter at middle latitudes and identified changes in the wind due to solar activity. The eastward component would reach 30-40 m/s during solar maximum, but around 23 m/s during low solar activity. On the other hand, the meridional component shifted from 0 m/s to 15 m/s in the southward direction during solar maximum and minimum, respectively. Later on, Bremer et al. (1997) identified weakly negative correlations with solar activity during most months (1964-1994) in the zonal component but with low significance, although the authors identified significant non-solar trends. Jacobi (1998) identified weaker eastward winds during solar maximum between 1972 and 1996. Portnyagin et al. (2006) studied the annual winds at middle latitudes between 1964 and 2004 and reported that zonal winds exhibited a decreasing trend while meridional winds increased until 1980. However, after this time period, no significant trends were observed. The authors also identified different trends for the summer winds, an increase in the summer zonal component in the 90s, as well as in the summer meridional component in the 70s, and concluded that these trends are non-uniform. Keuer et al. (2007) also found a correlation between solar activity and the MLT winds during summer and the trend shows an increase in the zonal wind and a decrease in the meridional component during 1990-2005. Later on, Jacobi et al. (2015) reported an increase with weaker tendencies in the eastward winds with a decrease in the southward component (1979-2014).

Hoffmann et al. (2010) compared one year of measurements from radars with the Kühlungsborn Mechanistic General Circulation Model (KMCM), showing the role of gravity wave drag in the summer MLT and the differences between middle and high latitudes and the interaction with waves between 12 h and 72 h. Later on, Hoffmann et al. (2011) studied the long-term behavior of the winds and gravity wave activity from kinetic energy over Germany and Norway and showed differences between the intensity of these two. Particularly, they found trends in the westward wind increasing around 75 km and a corresponding increase in gravity wave activity of 3-6 h above 80 km (1990-2010) at middle latitudes, while this was not the behavior observed at high latitudes (Hibbins et al., 2007). Offermann et al. (2011) also identified trends in the eastward wind due to an increase in gravity waves at 87 km with OH measurements.

Considering all the mentioned studies, the MLT exhibits varying trends over time, with distinct behavior during winter and summer due to differences in the seasonal wind dynamics inherent to the wind field properties. Additionally, many studies have focused solely on wind velocities at fixed altitudes. However, as mentioned before, research suggests that the MLT height has been decreasing over the past decades (e.g., Peters et al., 2017; Vincent et al., 2019; Yuan et al., 2019; Dawkins et al., 2023). In light of this, the present study examines the maximum velocity of the horizontal winds independent of their altitude, their variability, and trends over 19 years at high and middle latitudes over Europe during summer. In addition, the mesospheric time series at middle latitudes is extended to 33 years. Therefore, an introduction to the radar system and analysis methods implemented to extract the time series and analyze the trends is in Section 2. Section 3 describes the results obtained, while Sections 4 and 5 provide the discussion and concluding remarks, respectively.

## 2 Instruments and methods

### 2.1 Radar observations

The observational data used in this work is entirely from two types of radars: partial reflection radars (PRRs, also called MF radars) and specular meteor radars (SMRs). The PRR typically covers between 60 and 90 km altitude. Saura (69.14° N, 16.02° E) is located on Andøya, Norway, and has been in operation since 2004. This particular system operates with a peak power of 116 kW at a frequency of 3.17 MHz, having a Mills-Cross array that is composed of 29 crossed half-wave dipoles and thus offers a narrow beam for measurements (more information in Singer et al., 2005; Renkwitz and Latteck, 2017). A slightly smaller system which also has a Mills-cross shape is located in Juliusruh (54.63° N 13.37° E), Germany having an altitude coverage between 60 and 90 km. It has only 13 antennas and operates at a frequency of 3.18 MHz with a peak power of 64 kW. Starting as a Frequency Modulated Continuous Wave radar in 1990, it was modernized into a modular pulsed system (more details on the Juliusruh PRR systems can be found in Keuer et al., 2007; Singer et al., 1992).

The SMRs use the plasma trails left by meteors disintegrating in the atmosphere to retrieve the MLT winds by measuring their position and Doppler shift (e.g., Hocking et al., 2001). These systems are capable of measuring winds between 70 and 110 km (depending on the number of meteor detections). Particularly for this work, we have combined detections from two closely-located SMRs. This combination allows us to estimate the MLT mean winds reducing data gaps and improving the precision and continuity of the time series. The latter is especially useful for long-term studies (e.g., Jaen et al., 2022).

At high latitudes, the Andenes SMR (69.27° N, 16.04° E) and Tromsø SMR (69.58° N, 19.22° E), are combined for measurements between 2004 and 2022, with a 4 km-4 hr resolution to derive winds from 70 up to 110 km. In the case of middle latitudes, winds with 1 km-1 hr resolution have been obtained by combining meteor detections from Collm SMR (51.3° N, 13.0° E) and Juliusruh SMR (54.63° N 13.37° E), both operating in a pulsed mode. Note that both Andenes and Juliusruh SMR systems were upgraded in 2021 to operate in a coded continuous wave (CW) and Multiple Input Multiple Output mode (e.g., Huyghebaert et al., 2022; Poblet et al., 2023, for details of the upgrades in Andenes and Juliusruh). In this work, measurements from only one receiving station located close to each coded-CW transmitter are used.

The SMR systems cover a volume spanning from a 250-kilometer radius around a single system to 500 kilometers or more, depending on the number of systems in the network. The winds used in the study are representative of the entire volume covered by the systems. While the SMR winds measure a larger volume than the PRR winds, the mean zonal wind is not affected by this difference in volume. However, the meridional wind presents a relatively larger latitudinal and longitudinal dependence and these differences are more important. It has been studied and debated that PRRs underestimate the wind velocity above 80 km (Hall et al., 2005; Reid et al., 2018; Nozawa et al., 2002; Jacobi et al., 2009). Particularly for Saura PRR the winds are corrected based on the angle of arrival statistics and compared to mesospheric VHF wind measurements (Renkwitz et al., 2018).

## 2.2 Data analysis

In this study, most of the measurements have a length of 19 years (2004-2022), except for Juliusruh PRR with 33 years (1990-2022) at middle latitudes, from which we studied the mesospheric westward jet. To study the long-term behavior of the summer winds and their variability, we focus on the maximum median velocity per month as a proxy of the MLT dynamics. The different altitude ranges in the zonal and meridional data used for the climatologies aim to capture the maximum wind velocity. The zonal component is built with the combination of two datasets from different instruments while the meridional component is only from SMRs since it captures the maximum wind velocity during summer. To obtain the time series, we calculated monthly median values and extracted the maximum velocity between a range of altitudes corresponding to the peak and latitude, (i.e. westward jet 65-96 km, southward wind 75-95 km, and eastward jet 80-106 km). With the maximum wind velocity per month $v$, we implemented a linear function to fit by least squares $v = m \cdot yr + b$, where $m$ is the slope, $yr$ is the year and $b$ is the $v$-intercept. To test the slope of the linear fit we implemented the Student's t-test to reject the null hypothesis ($H_o : m = 0$) and calculated the confidence interval with the 95% confidence for the slope.

In order to study the variability of the time series, we implemented a Generalized Lomb-Scargle (LS) periodogram analysis with the difference between the 75% and 25% quartiles (third minus first) as an indication of the variability (since they are bigger than the measurements uncertainties) taken as the signal error (Czesla et al., 2019). The periodograms give the periods in years and normalized power provided by PyAstronomy (Zechmeister and Kürster, 2009). With this implementation, it is possible to obtain the False Alarm Probability (FAP) that responds to the questions of *"What is the probability that a signal with no periodic component would lead to a peak of this magnitude?"* over the highest peak, but it does not give information on the remaining peaks (VanderPlas, 2018). Even though the data is evenly spaced, one of the time series has a missing year (2000 in the Julisusruh PRR), which presents a complication in implementing the classical Fourier transform. In addition, Mossad et al. (2023) compared the Fourier transform to LS and found that LS is slightly more accurate for estimating the amplitude of a single frequency in the presence of minor gaps. The only disadvantage is the computation time, which in this study is not of importance given the number of data points.

It is widely known that there are many indices to categorize the atmosphere's external or internal forcing. In this case, we use the daily Ap index calculated from a network of magnetic observatories around the world. The Ap index varies between 0 and 400 and is the product of a conversion of the daily average of the 3-hour-mean Kp index (Matzka et al., 2021). Following the study at middle latitudes by Jacobi et al. (2021), we extend the work to 33 years below 82 km at middle latitudes, and to high latitudes with the 19-year time series and investigate the response to disturbed and undisturbed geomagnetic conditions during summer. We divided the days of the years with low geomagnetic activity, Ap $\leq$ 5, and high geomagnetic activity, Ap$\geq$ 20 for middle latitudes and Ap$\geq$ 15 for high latitudes. The reason for the distinction comes from the nature of the behavior of the geomagnetic field with the change of latitudes. Juliusruh is located at 52°N geomagnetic latitude, while Andenes is located at 67°N. Renkwitz and Latteck (2017) showed that the majority of the particle precipitation events already occur at Kp=3 ( Ap$\geq$ 15) which allows us to have a more robust time series considering the low geomagnetic activity in the last 19 years. In the case of middle latitudes, we use the limits already established by Jacobi et al. (2021). The time period used for the summer mean

is 2004-2022, except for the zonal vertical profile (below 82 km) and meridional (below 80 km) at middle latitudes, where the time period used is 1990-2022. Considering this selection, the total number of days between 2004 and 2022 is 888, 171, and 115 for Ap $\leq 5$, Ap$\geq 15$ and Ap$\geq 20$, respectively. In the case of 1990-2022, it is 1228 (Ap $\leq 5$) and 355 days (Ap$\geq 20$).

The summer mean vertical wind profile is determined by utilizing the time series from the selected Ap index. The difference between high and low geomagnetic activity in the summer winds is computed and then a Behrens-Fisher Student's $t$-test is calculated since the variance hypothesis is not satisfied (i.e. the variance of the samples are not assumed to be equal) and a combined degree of freedom is calculated for this objective (Robinson, 1976).

## 3 Results

### 3.1 Seasonal variations of winds

Figures 1a and 1b depict the climatologies of the mean-zonal winds between 60 and 110 km at middle and high latitudes, respectively. The climatologies are the mean of all the years (2004-2021) after a 16-day smoothing window shifted by 1-day. The horizontal lines at 79.5 km at middle latitudes and 85.5 km at high latitudes indicate the transition between the SMR and PRR measurements. Colors represent wind direction and intensity, while contour lines indicate wind velocity levels. Between January and March, the mean winds remain eastward until the springtime when the wind reversal occurs and the summertime begins (Jaen et al., 2022). During the summer months, the vertical wind profile (60-100 km) depicts the formation of the summer wind jets with an increase in the wind velocity in May and reaching the maximum velocities between June and August (see Figs. 1a and 1b in blue). As a result of the eastward wind in the lower thermosphere and the westward wind in the mesosphere, a strong wind shear around 83-86 km at middle latitudes (87-90 km at high latitudes) is located at the mesopause. The intensity of the wind jets differs quantitatively, at middle latitudes they are stronger than at high latitudes due to the mesospheric wind circulation. Below the zonal wind shear height, and between 72 and 76 km (75 and 78 km), the westward wind velocity maxima reach a mean of approximately 54 m/s (45 m/s). Above the strong wind shear and in the range of 93-98 km (97-100 km) the eastward jet mean is approximately 40 m/s (32 m/s). As August progresses the maximum velocity of the wind reduces and by the end of the summer (middle of September) the wind reversal occurs below 85 km (88 km) leaving eastward wind during the winter in the MLT (Jaen et al., 2022).

The meridional wind climatologies, Figures 1c and 1d were obtained equally as for the zonal component with the difference of using only the SMRs since the maximum summer wind altitude range is well captured by these radars. The meridional wind is less intense than the zonal wind. The velocity is quite variable in the observed range of 75-100 km. The velocity during the winter remains in the range of -5 to 5 ms$^{-1}$. The time period between June and July depicts the strongest southward wind throughout the year and is located between 82-86 km at middle latitudes (85-89 km at high latitudes) with medians of 16 m/s (13 m/s).

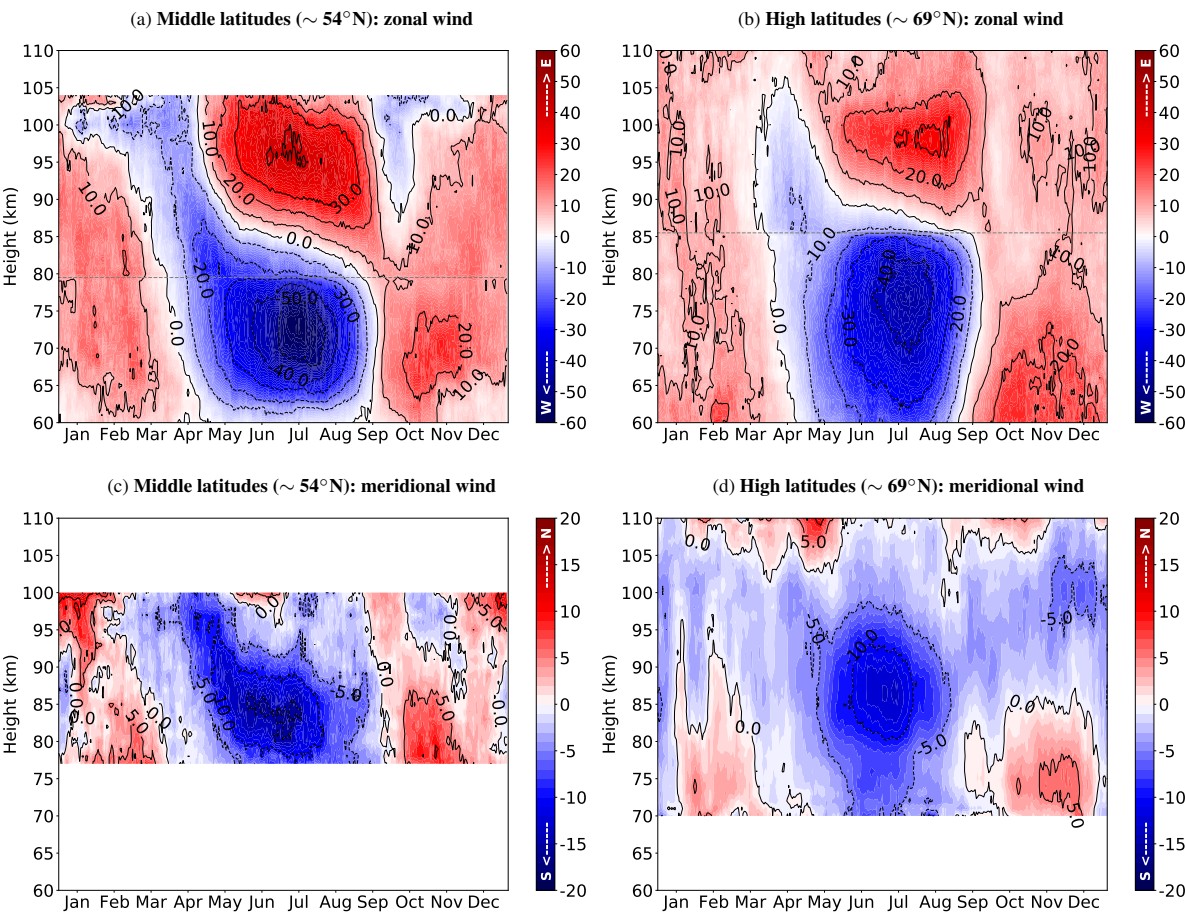

**Figure 1.** Horizontal wind height-time cross-section of the annual variation at middle (left column) and high (right column) latitudes. The upper row depicts the zonal (a, b) component with eastward (red) and westward (blue) wind velocity (m/s) by the color bar and the contour lines. The horizontal grey lines mark the change in the instrument. Similarly, the bottom row depicts the meridional component (c, d) with the northward (red) and southward (blue) wind velocity. Note that the meridional wind climatologies are only from SMR.

## 3.2 Trend in the horizontal winds

Figure 2 shows the time series peak velocity of the wind. Figures 2a and 2b are the eastward lower thermospheric jets per year for June, July and August (blue stars, red dots and green triangles) at middle and high latitudes, respectively. For each time series, the shaded area represents the interval between the first and the third quartiles (25th and 75th empirical quartiles) and the linear fit is displayed in the corresponding color. Through the implementation of the Student t-test and rejecting the null hypothesis (i.e. null slope, as discussed in Section 2.2), we obtain a statistical p-value. The color lines are the possible trends where m is the slope, wherein a dashed color line indicates a significance level exceeding 95%. Conversely, a dotted line suggests that the Student t-test did not reject the null hypothesis, implying that the slope could potentially be zero and

thus no significant trend exists. As a summary, the median height, the median velocity of the wind maxima, the slopes and the 95% confidence interval for the individual fit are listed in Table 1. In addition, the slopes with more than 95% significance are highlighted in bold.

| Wind proxy | Latitude (° N), years | Month | Height (km) | Velocity (m/s) | Slope (m/s yr$^{-1}$) | 95% confidence interval |
|---|---|---|---|---|---|---|
| Eastward | High (69) | June | $99 \pm 2$ | $25 \pm 2$ | $-0.12 \pm 0.21$ | ( -0.54, 0.30) |
|  | 2004-2022 | July | $98 \pm 2$ | $30 \pm 3$ | $\mathbf{-0.45 \pm 0.18}$ | (-0.81, -0.09) |
|  |  | August | $99 \pm 2$ | $32 \pm 3$ | $-0.28 \pm 0.19$ | (-0.68, 0.11) |
|  | Middle (53) | June | $97 \pm 1$ | $39 \pm 2$ | $-0.25 \pm 0.17$ | (-0.66 , 0.16 ) |
|  | 2004-2022 | July | $95 \pm 1$ | $40 \pm 1$ | $0.15 \pm 0.12$ | (-0.27, 0.56) |
|  |  | August | $94 \pm 1$ | $38 \pm 2$ | $-0.02 \pm 0.12$ | (-0.45, 0.40) |
| Westward | High (69) | June | $76 \pm 2$ | $-37 \pm 2$ | $0.02 \pm 0.16$ | (-0.40, 0.44) |
|  | 2004-2022 | July | $77 \pm 1$ | $-45 \pm 2$ | $-0.08 \pm 0.21$ | (-0.50, 0.34) |
|  |  | August | $77 \pm 3$ | $-39 \pm 2$ | $0.06 \pm 0.15$ | (-0.36, 0.48) |
|  | Middle (54) | June | $74 \pm 1$ | $-43 \pm 4$ | $\mathbf{-0.39 \pm 0.10}$ | (-0.64, -0.14) |
|  | 1990-2022 | July | $74 \pm 2$ | $-54 \pm 6$ | $\mathbf{-0.64 \pm 0.10}$ | (-0.84, -0.44) |
|  |  | August | $74 \pm 1$ | $-35 \pm 6$ | $\mathbf{-0.41 \pm 0.09}$ | (-0.64, -0.17) |
| Southward | High (69) | June | $88 \pm 2$ | $-12 \pm 2$ | $0.05 \pm 0.09$ | ( -0.36, 0.47) |
|  | 2004-2022 | July | $87 \pm 2$ | $-13 \pm 1$ | $-0.06 \pm 0.08$ | (-0.48, 0.35) |
|  |  | August | $86 \pm 2$ | $-9 \pm 1$ | $-0.02 \pm 0.06$ | ( -0.44, 0.40) |
|  | Middle (53) | June | $84 \pm 2$ | $-16 \pm 2$ | $\mathbf{0.33 \pm 0.12}$ | (-0.03, 0.70) |
|  | 2004-2022 | July | $82 \pm 1$ | $-16 \pm 2$ | $\mathbf{0.26 \pm 0.12}$ | (-0.13, 0.65) |
|  |  | August | $85 \pm 1$ | $-7 \pm 2$ | $\mathbf{0.23 \pm 0.10}$ | (-0.13, 0.60) |

**Table 1.** Summary of the wind maxima characteristics with the respective latitude, the range of years, median height, median velocity, monthly slope with standard deviation, and the 95% confidence interval. Highlighted in bold, are the significant trends with more than 95% confidence.

The eastward jets at middle latitudes (Fig. 2a) show no significant trends. However, at high latitudes, July depicts a significant trend with more than 98% (see Fig. 2b), indicating weaker eastward winds over the years. In the case of the mesospheric westward jet at high latitudes, the Student t-test does not reject the null slope hypothesis, but the westward jet at middle latitudes depicts significant trends with more than 99.9% for all the months, indicating a tendency of stronger westward winds since 1990 (Fig. 2c).

Figures 2e and 2f depict the southward winds at middle and high latitudes, respectively. In both cases, the jets are stronger during June and July than during August. While the southward maximum wind velocity remains approximately constant at high

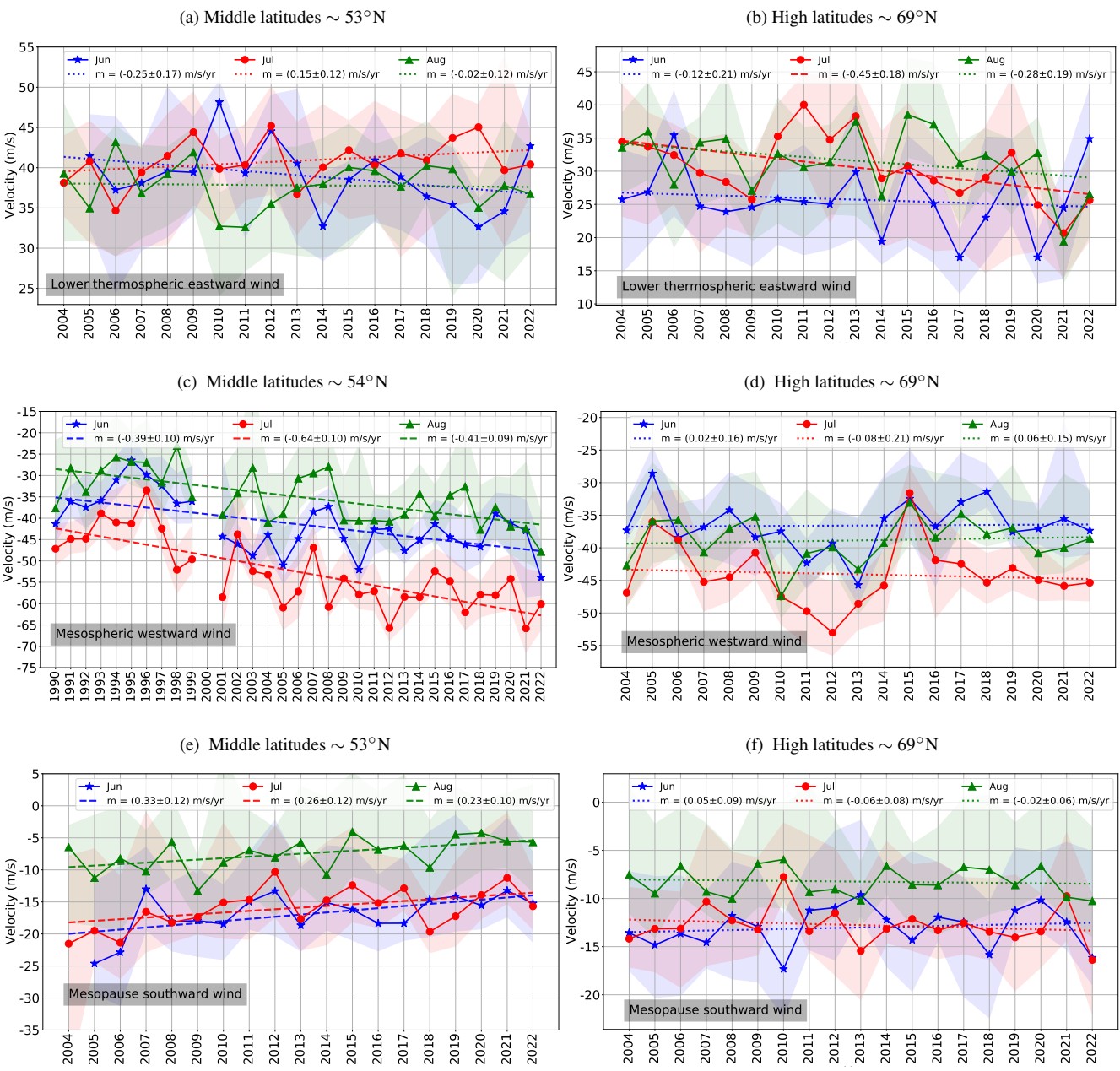

**Figure 2.** Middle (left column) and high (right column) latitudes zonal and meridional wind maxima for every year. The eastward (upper row), westward (middle row) and southward (bottom row) velocity maxima. Each wind component has the yearly velocity maxima obtained with a monthly median and their respective quartile difference (i.e. 75 and 25 quartiles). June (blue stars), July (red dots), and August (green triangles) with the linear fit where $m$ represents the slope. The dashed lines depict the trends with 95% significance.

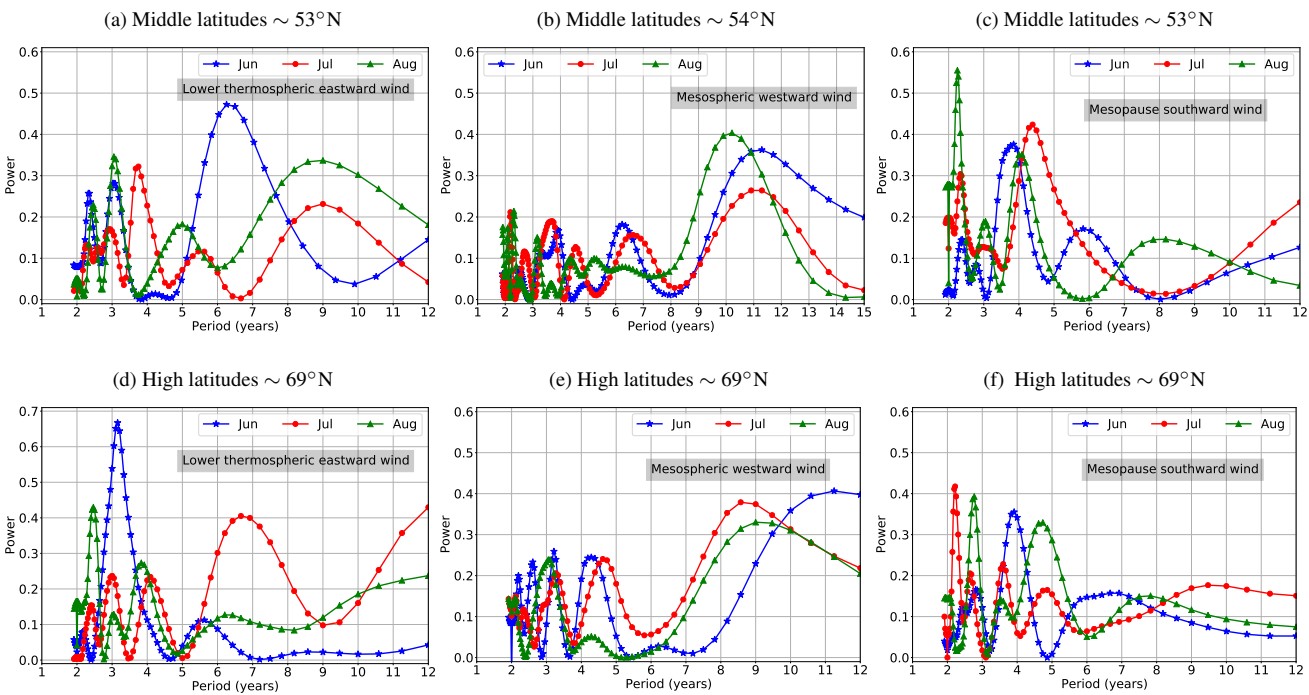

**Figure 3.** Middle (a, b, c) and high latitudes (d, e, f) periodograms calculated with Lomb-Scargle. The left column are the periodograms from the timeseries of the eastward maxima (a, d), the middle are the westward maxima (b, e) and the right column is the southward winds maxima.

latitudes, at middle latitudes a significant trend (more than 95% confidence) is visible indicating a weakening of the meridional wind component over the years.

The altitude of the wind maxima fluctuates from year to year while consistently maintains its mean height (see Table 1) without any significant trends between 2004 and 2022.

### 3.3 Inter-annual variability of winds

The time series in Figure 2 reveals year-to-year variability, which motivated us to investigate the interannual variability of the time series through periodogram analyses. Figure 3 depicts the Lomb-Scargle periodograms of winds for each summer month. The upper row corresponds to middle latitudes, while the bottom row refers to high latitudes. The columns from left to right represent the lower thermospheric eastward, mesospheric westward, and mesopause southward winds, respectively

Table 2 summarizes the corresponding periods. At high latitudes, the eastward wind exhibits significant (over 90% confidence level) periodicities of 2-3 years in June and August, and of 12 years in July. At middle latitudes, significant periodicities are seen in the eastward wind around 6 years in June, in the westward wind around 10-11 years in June and August, and 2-4 years in the southward wind in July and August.

|  |  | Periods (years) | | |
| Latitude (°N) | Wind (mean height) | June | July | August |
| --- | --- | --- | --- | --- |
| High (∼69) | Eastward (99 km) | **3.2**\* | 6.7, **12.0** | **2.5**, 3.8 |
|  | Westward (76 km) | 4.3, 11.3 | 3.3, 4.6, 8.6 | 3.1, 9.0 |
|  | Southward (87 km) | 3.9 | 2.2, 3.6 | 2.8, 4.7 |
| Middle (∼54) | Eastward (95 km) | 2.3, 3.1, **6.3** | 3.8, 9.0 | 2.5, 3.1, 9.0 |
|  | Westward (74 km) | **11.3**\* | 11.3 | **10.2**\* |
|  | Southward (84 km) | 3.8 | 2.3, **4.4** | **2.3**\*, 4.1 |

The periods in bold are the ones that passed the false alarm probability of 90% and the ones with a star the 95%

**Table 2.** Summary with the periods obtained from LS analysis of monthly time series.

### 3.4 Wind response to geomagnetic activity

Figure 4 shows the summer wind at high latitudes under quiet and disturbed geomagnetic conditions over the years. Figures 4a and 4b depict the yearly median summer zonal wind at low (Ap $\leq$ 5) and high (Ap $\geq$ 15) geomagnetic activity, respectively. On a simple visual examination, an enhancement in the eastward jet (red) under high geomagnetic activity is evident, while the westward jet (blue) remains in the same velocity range. The meridional component under disturbed geomagnetic conditions (Fig. 4d) displays a more variable velocity than at low geomagnetic activity (Fig. 4c). Note that for high geomagnetic activity (Figs. 4b and 4d), the year 2020 depicts significant differences with the rest of the years due to the reduced number of days (2) with Ap $\geq$ 15.

In order to quantify the possible differences, a summer mean with its standard deviation is calculated. The altitude velocity profiles for the summer mean at high latitudes are shown in Figure 5 with low (green) and high (purple) geomagnetic activity for the zonal component (Fig. 5a) and the meridional component (Fig. 5b). They show stronger eastward winds under high geomagnetic activity above 92 km, while the rest of the profile does not exhibit a distinct difference between high and low geomagnetic activity. In the case of the meridional component (Fig. 5b), considerable difference appears below 85 km, with stronger northward wind under quiet geomagnetic conditions. Figure 5c shows the difference between low and high geomagnetic activity for the zonal (red) and meridional (blue) summer wind components. Significant mean differences beyond 95% according to the Behrens-Fisher test are denoted by stars, while circles indicate differences where the hypothesis of equal means was not rejected by the test. The summer eastward wind is significantly affected by geomagnetic activity with 2-6 m/s stronger eastward wind velocities above 94 km and 1-3.5 m/s weaker northward wind below 83 km for strong geomagnetic activity.

A similar analysis is done for mid-latitudes, with the results shown in Figure 6. Note that the zonal wind between 70-82 km and meridional wind 70-80 km are obtained between 1990 and 2022, while the winds above are obtained from 2004-2022

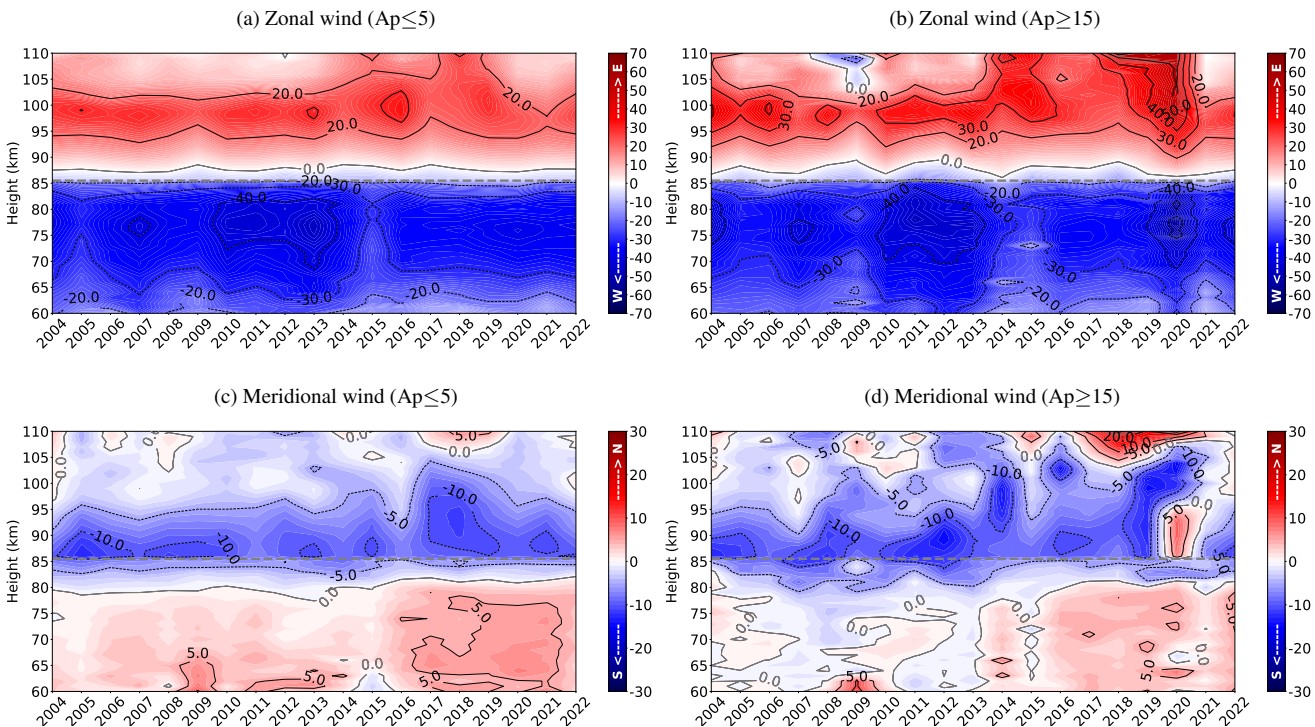

**Figure 4.** Summer zonal winds at high latitudes over the years with low (a) and high (b) geomagnetic activity. In the same way, the bottom row depicts the summer meridional mean winds with low (c) and high (d) geomagnetic activity, also for high latitudes.

due to the use of different systems (see Section 2). The selection of the different range of altitudes/systems over the wind components is to avoid the instrument shift over the wind maxima, the focus of our study.

As shown in Figure 6c, higher geomagnetic activity has a significant impact on the middle latitudes zonal wind, decelerating both the eastward jet (by up to -10 m/s) above 95 km altitude and the westward jet below 80 km (by up to 8 m/s). Its impact on the meridional wind is mainly seen below 78 km altitude, decelerating the northward wind by up to -3 m/s. Note that there are differences between the meridional wind from Figures 1d and 4(c, d), which are a consequence of different instruments (see Section 2.1). Table 3 contains the trends calculated for July eastward and the summer westward maximum at high and middle latitudes, respectively with only the days of low geomagnetic activity to compare with the significant trends in the zonal wind.

## 4   Discussion

In this work, we have explored the long-term variability of MLT summer wind using the maximum wind velocity as a proxy. The resulting time series were fitted with linear functions and the slopes were tested in search of significant trends. In addition, we calculated the periodograms of the time series and explored the response of the wind under disturbed and non-disturbed geomagnetic conditions at high and middle latitudes.

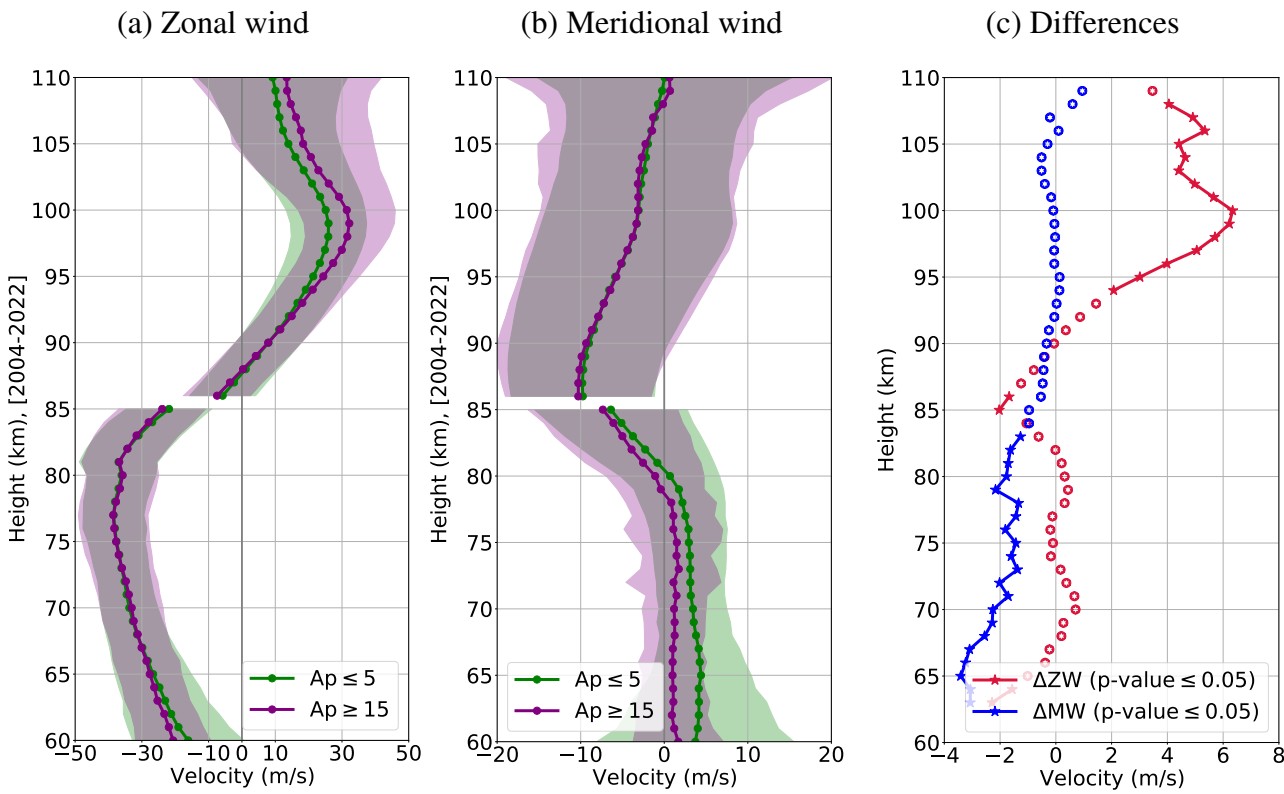

**Figure 5.** Mean velocity profiles at high latitudes for low (green, Ap $\leq$ 5) and high (purple, Ap $\geq$ 15) geomagnetic activity for the summer zonal (a) and meridional (b) winds. The difference between both profiles under high and low geomagnetic activity (c) for the zonal (red) and the meridional (blue) wind component. The stars depict the values with more than 95% significance, tested with the Behrens-Fisher test and the circles the values with no significant difference between the means.

| Latitude | Wind proxy | Slope (m/s/y) |
|---|---|---|
| ∼69°N | July eastward (99 km) | **-0.44 ± 0.15** |
| ∼53°N | Summer westward (74 km) | **-0.41 ± 0.08** |

**Table 3.** Wind maxima slopes obtained with the linear fit with days of low geomagnetic activity. Highlighted in bold, are the significant trends with more than 95% confidence.

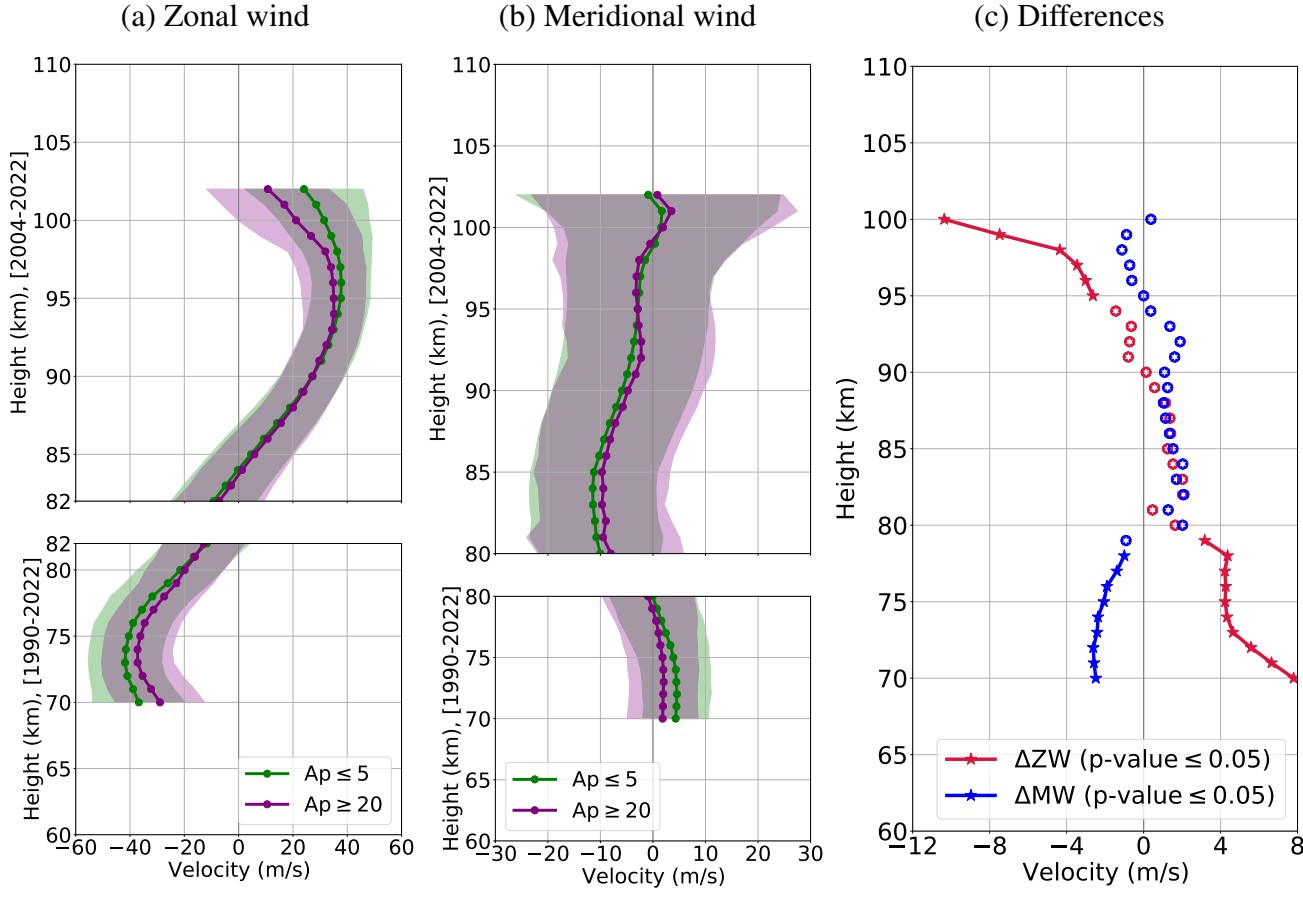

**Figure 6.** Mean velocity profiles at middle latitudes for low (green, Ap ≤ 5) and high (purple, Ap ≥ 20) geomagnetic activity for the summer zonal (a) and meridional (b) winds. In the range of 70-82 (80) km, the zonal (meridional) components are calculated for the years 1990-2022. Above these heights, the analyzed years are 2004-2022. (c) The difference between both profiles under high and low geomagnetic activity for the zonal (red) and the meridional (blue) wind components. The stars depict the values with more than 95% significance, tested with the Behrens-Fisher test, and the circles are the values with no significant difference between the means.

### 4.1 Seasonal wind variations and trends

The wind climatologies are in agreement with previous studies at similar latitudes, considering the difference of height and investigated time lengths (Wilhelm et al., 2019; Hoffmann et al., 2010; Jaen et al., 2022; Schminder et al., 1997; Manson et al., 2004, etc.). However, a comparison with models shows that in the summer season, WACCM-X(SD) and UA-ICON exhibit a good agreement with radars, while winter is better represented by GAIA (Stober et al., 2021). Accordingly, Zhou et al. (2022) compared a network of meteor radars located at different latitudes (from middle to low latitudes) with WACCM-X(SD) finding similar results.

The wind trends observed in July for the lower thermospheric eastward maxima at high latitudes and the southward maxima at middle latitudes are consistent with the study made by Wilhelm et al. (2019). In their work, the authors studied the trends at high latitudes and middle latitudes from 2002 to 2018 with specular meteor radars, and with this limitation, the westward wind maxima are not captured in their study, although a significant westward wind trend is visible below 85 km during the summer months. Hall and Tsutsumi (2013) made a similar study comparing two SMR at latitudes of 70°N and 78°N between 2001 and 2012. The authors identified a strengthening of the summer westward jet contradicting our results. However, considering the time period that their results overlap our study (i.e 2004-2012), it is visible in Figure 2d that a possible significant trend could be found. On the other hand, Jacobi et al. (2023) analyzed 43 years of the winds near 90 km and 81–85 km over Collm and Juliusruh, respectively, obtaining significant trends at Juliusruh in the summer month eastward wind that do not agree with our findings. The differences can be attributed to the varying heights that were studied, which do not capture the proxies from this study. Hoffmann et al. (2011) studied the trends in the zonal wind for July at middle latitudes and found significant trends at 72 km and 76 km where the lower and upper limit of the westward jet is located. These trends showed stronger westward winds of 1.1 m/s/year and 0.643 m/s/year, respectively, over the period of 1990-2010. Vincent et al. (2019) studied the trends in the meridional wind over the Antarctic (∼69°S) between 1994 and 2018 during the Austral summer. While the study shows a descent in the height of the maximum wind velocity by 1.5 km, the strengths of the wind maxima did not change, which agrees with our findings in the northern hemisphere.

It is essential to highlight that while the zonal wind shows a good representation of the global zonal wind behavior, the meridional component has a higher dependence on the region where it is observed, as a consequence of the gravity wave input at the mesopause (e.g., Jacobi et al., 2001). This could also explain the variations in trends across different latitudes and longitudes, as well as the limitations of global models, which often rely on parametrizations. Moreover, different lengths of the time series, their locations and parameters may show different trends due to the complex dynamics. Laštovička and Jelínek (2019) listed the main difficulties when studying trends.

### 4.2 Inter-annual oscillations

From the periodograms, several periods are identified (Fig. 3). Those periods of around 2-3 years could be associated with modulations from the stratospheric QBO, also called mesospheric QBO (MQBO). Espy et al. (2011) showed QBO modulation on the summer mesospheric OH temperatures at 60°N. The mesopause southward winds are a result of the eastward phase

gravity waves that reach the mesopause where they break and deposit momentum. Coupled with the Coriolis force, the meridional component of these winds drives the summer-to-winter pole circulation (e.g., Andrews et al., 1987; Holton, 2004). These gravity waves are previously filtered by the stratospheric flow, which is predominantly influenced by QBO (e.g., Baldwin et al., 2001; Lindzen and Holton, 1968). As such, these filtered gravity waves may be the underlying cause of the observed wind oscillations in the mesopause.

The periods between 3-4 years have been associated with modulation from ENSO. Reid et al. (2014) obtained oscillations of 3-4 years in OH and O(1s) airglow at 96 and 87 km $\sim$35°S. Perminov et al. (2018) obtained similar oscillation in OH temperatures 56°N (2000-2016). García-Herrera et al. (2006) found a lag of a few months between ENSO and the temperature response in the stratosphere at high latitudes. Considering that ENSO's main phase occurs in December and it is an ocean-atmospheric event at equatorial latitudes, a signal's attenuation with time and space would be expected. While these findings are a result of mesospheric temperature observations, they are intrinsically linked to the mesospheric dynamics as expressed in the thermal wind equation. Jacobi and Kürschner (2002) identified a possible signature of ENSO with Collm zonal winds in the 1980s and 1990s, and later on, Jacobi et al. (2017) found that this signal changes between the mesosphere and lower thermosphere. However, having common periodicity does not necessarily mean causality and dedicated work needs to be done to connect these oscillations to QBO and ENSO phenomena.

### 4.3 Solar cycle dependence

The summer wind also shows a significant periodicity around 6 years and 10-12 years, which could be a signature of the solar cycle and its harmonic (see Table 2). Previous works have shown connections between the solar cycle and the winds (e.g., Jacobi and Kürschner, 2006), while other authors have shown these signatures have disappeared in past solar cycles (Portnyagin et al., 2006; Fiedler et al., 2011; DeLand and Thomas, 2015). For our data, a simple Pearson correlation between the Lyman-$\alpha$ and the June westward wind maximum at middle latitudes gives a $\rho = -0.11$ with a $p$-value of 0.56, which indicates a non-significant anti-correlation. When considering the periods of 1990-2004 and 2004-2022 separately, we find two different correlations of $\rho = -0.69$ ($p$-value = 0.01) and $\rho = -0.17$ ($p$-value= 0.51), respectively. The first one shows a significant ($\geq 95\%$) anti-correlation between the westward wind maximum and Lyman-$\alpha$ during the time length of 1990 to 2004, while the latter is not significant. Recently, Vellalassery et al. (2023) showed how the attenuation of the solar cycle response with Lyman-$\alpha$ is correlated with the increase of water vapor in the mesosphere at polar latitudes.

### 4.4 Impact of geomagnetic activity on trends

Our results reveal an impact of geomagnetic activity on MLT winds in summer (Figures 5 and 6), with higher geomagnetic activity weakening the winds at both middle and high latitudes, except for the eastward wind at high latitudes, which strengthens under disturbed conditions. The results at middle latitudes agree with Jacobi et al. (2021), who found weaker wind at a higher geomagnetic activity at two mid-latitude stations Collm and Kazan during the summers of 2016-2020.

Li et al. (2023) and Li et al. (2019) studied the response of the MLT during a geomagnetic storm with the Thermosphere Ionosphere Mesosphere Electrodynamic General Circulation Model (TIMEGCM) at high and middle latitudes, respectively.

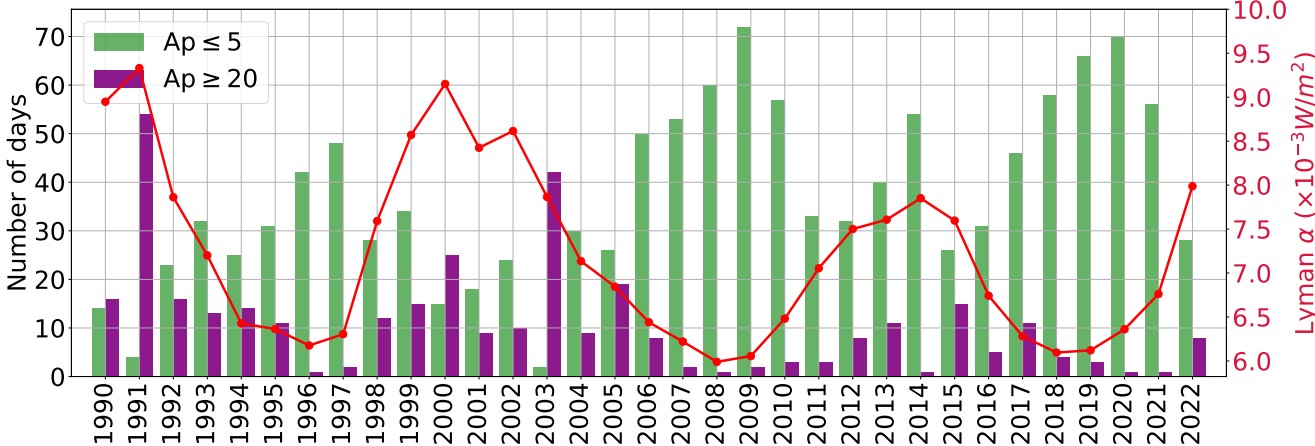

**Figure 7.** Histogram with the counts of days for Ap ≤ 5 (green) and Ap ≥ 20 (purple). On the right axis is the Lyman α (red) scale with yearly summer mean values.

While the effects of Joule heating penetrate up to ∼95 km at 70°N, the temperature contribution is minor compared to vertical heat advection and the adiabatic processes. Above 100 km the Joule heating becomes significant and comparable to the other two processes. In the MLT the major heating contributions are due to adiabatic heating/cooling and vertical heat advection associated with vertical winds. During a storm, the temperature increases and decreases with the vertical wind advection, and the horizontal advection also contributes to the total heating rate (Li et al., 2023). The meridional winds showed a shift in the direction, from southward to northward, similar to the decrease in the winds observed below 79 km (Figs. 5b and 6b). These changes are a consequence of the temperature changes leading to downward/upward vertical winds which together with the Coriolis force produce a change in the pressure gradient (Li et al., 2019). However, Sun et al. (2022) compared the simulation with TIMEGCM and SABER and found that the storm effects penetrate down to ∼80 km and the model agrees with the observation, but overestimates the temperature increase and underestimates the temperature decrease at high and middle latitudes.

The geomagnetic activity could significantly affect long-term trend studies as demonstrated by Liu et al. (2021). Given the significant geomagnetic activity effect on the mesosphere wind, the long-term variation of the geomagnetic activity could potentially contribute to the wind trend we obtained in Figure 2. The histogram in Figure 7 shows the count of days from 1990 to 2022 with Ap ≤ 5 (green) and Ap ≥ 20 (purple). The right axis shows the scaled Lyman α (red line) to illustrate the solar activity over the investigated time period. Comparatively, the first half of the time series has more days with high geomagnetic activity than the second half. This could imply that the negative trend in the westward jet can partially be due to more quiet days in the recent two decades. To test this, we calculated the summer mean (June-August averaged) maximum velocity from the yearly summer winds under only quiet conditions. At high latitudes, the eastward jet shows a significant summer trend of -0.23 ± 0.13 m/s/yr and a July eastward trend of -0.44 ± 0.08 m/s/yr, in agreement with the complete time series (see Table 1).

At middle latitudes, the westward jet also shows a significant trend of -0.41 ± 0.08 m/s/yr which is similar to the summer mean with the complete time series (-0.49± 0.13 m/s/yr). This result suggests that the winds are getting stronger due to different causes.

As to the cause of the long-term trends in the winds observed here, it could be related to changes in various atmospheric waves, such as planetary waves, tides, and gravity waves as discussed by Laštovička et al. (2012). However, during summer the contribution of tides is filtered at lower altitudes in the stratosphere leaving less contribution in the mesosphere (Conte et al., 2017; Wilhelm et al., 2019; Pedatella et al., 2021). Ern et al. (2011) and Alexandre et al. (2021) found evidence of gravity waves generated in the lower latitudes of the summer hemisphere contribute to the momentum budget of the mesospheric jet in the 340 middle and high latitudes. Hoffmann et al. (2011) found a significant trend of increasing gravity waves above 80 km. Liu et al. (2017) studied the gravity wave potential energy derived from satellite-observed temperatures and also found a positive trend below 80 km during July at 50°N. While Luo et al. (2021) found a decreasing gravity wave trend in the stratosphere between 2007 and 2020 derived from temperature observations of GNSS radio occultation. A likely scenario could be a decrease in the stratospheric filtering of gravity waves, allowing them to reach higher altitudes and deposit momentum flux in the mean 345 westward flow (Medvedev et al., 1998; Yiğit and Medvedev, 2017; Conte et al., 2023). In a similar way, the southward wind is decreasing and this could be a consequence of less energy reaching the mesopause due to the intense westward flow. In the future, we want to explore this possibility to identify the origin of this long-term trend.

## 5   Concluding remarks

The current manuscript examines long-term variations by analyzing the median maximum wind velocity in June, July, and 350 August as an indicator of wind dynamics over the years. Linear functions were fitted to the time series, and the slope was tested using Student's t-test, in addition, Lomb-Scargle periodograms were calculated. This study also investigates the relationship between wind patterns and geomagnetic activity by using the Ap index and tests its influence on the identified trends. The results are summarized as follows:

– The lower thermospheric eastward summer maximum in July shows a significant trend from 2004 to 2022 at high latitudes, with a decrease of (0.45±0.18) m/s per year, exceeding 95% statistical significance. The wind velocity reaches its peak between June and August, ranging from 25 to 32 m/s at an altitude of 98-99 km.

  – The mesospheric westward summer maximum has strengthened over the past 33 years (1990-2022) at middle latitudes. The highest velocity occurs in July, where it exhibits a significant trend of 0.64 ± 0.10 m/s. The summer mesospheric westward trends are independent of the geomagnetic activity.

– The mesopause southward wind velocity experiences a significant decline during the three studied months at middle latitudes between 2004 and 2022, with the most substantial decrease occurring in June and July with slopes of 0.33 ± 0.12 and 0.26 ± 0.12 m/s per year, respectively.

- The summer mesospheric westward wind maxima exhibit an oscillation related to the solar cycle (8.6-11.3 years). This oscillation is significant for the months of June, July, and August during the period from 1990 to 2022. Additionally, other oscillations with periods of 2.3-2.8 years and 3.2-4.4 years are present in most of the time series and could be associated with modulation from the QBO or ENSO.

- Geomagnetic activity induces higher lower thermospheric eastward winds above 94 km at high latitudes and weaker zonal winds at middle latitudes above 95 km and below 79 km.

- The mesopause southward wind displays a disturbed pattern under high geomagnetic activity, reducing the wind velocity below 84 km at high latitudes and below 78 km at middle latitudes.

As the Earth's atmosphere continues evolving, the pursuit of long-term studies becomes increasingly challenging, yielding changing results over the years. Therefore, the acquisition of longer time series becomes imperative in order to truly comprehend the dynamics of the MLT. Although models have made notable improvements in their results, they still encounter certain limitations that need to be addressed. While measurements provide localized insights and show specific latitudinal characteristics, they inherently lack a comprehensive view of the entire system. Additionally, understanding the complexities of gravity waves in the middle atmosphere is crucial, as they emerge as a significant energy source in the MLT dynamics.

*Data availability.* The data to produce the figures is available in HDF5 format at DOI: 10.22000/1603. This link is temporary for the reviewing process. Once it is accepted, it will become permanent.
(https://www.radar-service.eu/radar/en/dataset/HOYslSXhNytHHPUf?token=TkzTyxslRwCelhXQwwAN)

*Author contributions.* JJ, TR, HL and JC developed the idea and helped in the interpretation of results. NG and MT ensured the operation of the Tromsø specular meteor radar and CJ of the Collm specular meteor radar. The writing of this paper was done by JJ with the assistance and contribution of all authors to the discussion, draft review, and editing.

*Competing interests.* The authors declare that they have no conflict of interest.

*Acknowledgements.* JJ thanks Axel Gabriel and Federico Conte for the helpful discussions and Matthias Clahsen for the data processing.
Lyman $\alpha$ is obtained from LISIRD (https://lasp.colorado.edu/lisird/), last access 18.02.2023. Ap index is available from GFZ (https://www.gfz-potsdam.de/en/section/geomagnetism/data-products-services/geomagnetic-kp-index), last access 25.11.2022.
This research has been supported by the Deutsche Forschungsgemeinschaft (grant nos. PO 2341/2-1 and JA 836/47-1) and the Bundesministerium für Bildung und Forschung (TIMA, grant no. 01 LG 1902A). HL acknowledges support by the JSPS KAKENHI grants 18H01270, 17KK0095, and JRPs-LEAD with DFG (JPJSJPR 20181602).

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
