# Peer review of "Long-term studies of the summer wind in the mesosphere and lower thermosphere at middle and high latitudes over Europe"

_EGUsphere, 2023_

## Referee Comment (RC2)

This study is focusing on the long-term trend of the horizontal wind in the MLT, using multiple radar datasets in high and middle latitudes. The calculation of the trends of horizontal wind is always challenging due to the large variability of the winds. These results will contribute greatly to the community's understanding on the MLT dynamics changes induced by increasing $CO_2$ and climate change in the lower atmosphere. In addition, the author analyzed the radar date during geomagnetic quiet time and active time and revealed the impacts of the geomagnetic activity on the horizontal wind. This is an important topic that has not been studied very much in the community. There are some recent first principle model investigations, using TIME-GCM, on the MLT wind responses to the geomagnetic storms, showing dramatic variations in the MLT wind field during geomagnetic activities (Li et al., 2019, 2023). They could be helpful for the understanding of these results and the discussion section in this paper. One the other hand, the algorithm is quite different to the traditional multi-linear regress approach. I encourage the author to conduct a separate multi-linear regression analysis to look at specifically the geomagnetic effects, and compare them with the current results. Because the wind results of high geomagnetic activity are based on considerably less number of days than those of low geomagnetic activity. It would be very interesting to see if they are consistent with each other. In addition, since there are two types of radar involved in this study (MF radar and MWR for zonal wind data), the author should be aware of the potential bias between the two instruments (MF radar underestimates the winds) and address the possible effects on the results presented. Reid et al (2018) has some insightful discussion on this topic, and the paper could be beneficial for the author to clarify this issue. There is not much investigations on the inter-annual oscillations and solar cycle effects in the current manuscript. I suggest removing 4.1 and 4.2, just focusing on wind trends and geomagnetic effects.

Minor issues:

The author keeps using the term "velocity amplitude" throughout the paper. This is confusing, since "amplitude" is usually referring to the magnitude of the modulation, such as wave amplitude, but this study is about the mean wind. I think this is due to some writing habit. Please revise.

Page 1, line 4: "…absence of intense planetary wave…". This is incorrect. The QTDW is quite active in the summer hemisphere. But it should not affect the results, since the author is using the sliding16-day window.

Page 3, line 78, The MLT height decrease is also revealed by Yuan et al., 2019 on the trend of the mesopause height, a direct evidence of "shrinking" of the upper atmosphere.

Page 5, the definition of low and high geomagnetic quiet days seems arbitrary. For high latitude AP >= 15, but for midlatitude AP>=20. I understand that you need stronger geomagnetic activity to see the changes at midlatitudes, but I think the criteria should be consistent. In addition, it is expected to see weaker responses at midlatitudes than at high latitudes.

Page 9, line 198-199, this is expected from the model simulations mentioned above. But the high latitude responses are more complex than those at the midlatitudes due to aurora heating.

Page 13, why leaves a gap in each of figure 6a and 6b?

Page 14, I suggest deleting the statement "In addition,…", because there is not much investigation on this topic, just some hypothesis. See my comment above.

Page 15, "… the contribution of planetary waves is ….." see my comment above.

Suggested references:

Li, J., Wei, G., Wang, W., Luo, Q., Lu, J., Tian, Y., Xiong, S., Sun, M., Shen, F., Yuan, T., Zhang, X., Fu, S., Li, Z., Zhang, H., Yang, C. (2023). A modeling study on the responses of the mesosphere and lower thermosphere (MLT) temperature to the initial and main phases of geomagnetic storms at high latitudes. Journal of Geophysical Research: Atmospheres, 128, e2022JD038348. https://doi.org/10.1029/2022JD038348

Li, J., W. Wang, J. Lu, J. Yue, A. Burns, T. Yuan, X. Chen and W. Dong (2019): A modeling study of the responses of mesosphere and lower thermosphere (MLT) winds to geomagnetic storms at middle latitudes, J. Geophys. Res. Space Physics 124. https://doi.org/10.1029/2019JA026533.

Reid, I.M., McIntosh, D.L., Murphy, D.J. et al. Mesospheric radar wind comparisons at high and middle southern latitudes. Earth Planets Space 70, 84 (2018). https://doi.org/10.1186/s40623-018-0861-1

Yuan, T., Solomon, S. C., She, C.-Y., Krueger, D. A., & Liu, H.-L. ( 2019). The long-term trends of nocturnal mesopause temperature and altitude revealed by Na lidar observations between 1990 and 2018 at mid-latitude. Journal of Geophysical Research: Atmospheres, 124. https://doi.org/10.1029/2018JD029828.

---

## Author Comment (AC1)

**Response to RC1**: , Anonymous Referee #1, 08 Aug 2023.

*We appreciate the valuable comments given by the reviewer. We repeat the reviewer's concerns and provide our respective responses in italics.*

In this work, the authors analyze radar wind datasets from two sites at different latitudes, infer summer wind maxima in wind components and derive trends. They find a robust strengthening of the mesospheric westward wind at mid latitudes. The influence of daily varying geomagnetic activity on the winds is studied and excluded as source for significant trends. Gravity waves are suggested as cause of the trends.

The paper is well structured and the figures are clear. In few places there are descriptions that can be confusing, mainly when the authors speak of "maximum velocity amplitudes" when actually absolute wind values are meant, which is likely a language problem. I notice that title is a bit unspecific, being almost identical to the special issue title.

*We understand the confusion with the nomenclature. When we observe a monthly median vertical profile of the horizontal velocities, the wind jets look like a wave with the maximum as the highest amplitude of the velocity. From there we started to call it amplitude. We will remove the 'amplitude' word in the manuscript.*

*The title has been changed to "Long-term studies of the summer wind in the mesosphere and lower thermosphere at middle and high latitudes over Europe".*

The authors list a number of similar observational studies, sometimes based on the same datasets, that sometimes come to contradicting results regarding the trends. This is attributed to different years or altitude ranges. If trends depend so sensitively on the selection of years or altitude ranges, this should be investigated in more detail to be of scientific value. For example, if altitude of wind maxima change with time, this could have been documented.

*Indeed there are studies indicating that the MLT heights have decreased in the past decades (Peters et al. 2017, Dawkins et al. 2023). Having this knowledge in mind, we used the wind maxima and not the altitude as the object of study. We did explore the altitudes of the wind maxima and found no trend in the 19-year time series. For the first years of the longer time series (33 years), the accuracy of measured altitudes of this initial radar system is arguable. So in the case of testing for a trend, we only consider the years after 2004. We haven't included the plots since there are no significant trends, but indeed they show variability over the years. Below are the Figures, in red colors are the eastward jets, in purple are the westward jets*

[Figure]

*and in blue are the southward wind maxima. The error bars are not included, but the distribution depends on the instruments as follows: Middle latitudes eastward and southward +- 1km, westward +- 5km. High latitudes eastward and southward +- 2km westward 2km.*

Another possible source of variability, longitude, is not mentioned at all. The authors generally refer to "middle and high latitudes" and "zonal mean winds" and therefore make the impression that their results are valid for all longitudes. This however cannot be inferred from the datasets the authors used, and some discussion of this topic should be included. Whether the obtained results are only valid locally or in the zonal mean is relevant.

*Thank you for highlighting this. Indeed our results are local since the radars are in a fixed location. We did mention this in section 2.1 lines 106-107. However it is known that the zonal climatologies are a good representation of global behavior, but the trends are not. This is one of the potential reasons why we have found strong trends while other authors and instruments at different locations don't. Furthermore, we suspect that the observed variability is actually a consequence of GWs or PWs, including their local/regional properties. We are including this comment in the discussion.*

No references to model studies were made, and models were mentioned only briefly in one sentence at the end of the manuscript. The work is mainly a report of measurements without detailed exploration of the findings regarding the underlying mechanisms. For example, periodograms with periods of 2-4 and 11 years are presented but discussion with references to QBO and ENSO remain very vague. The same applies to gravity waves being the suggested cause of the trends, but no proposals or attempts were made at how this could be tested or verified. If this is out of scope, but similar studies are available, that sometimes agree, sometimes contradict, the value of presenting measurements only is limited.

*We are adding citations of works using models in the discussion briefly. Sadly there are not many studies with models that represent the MLT summer well enough in order to compare it directly. We will add more citations. In the case of QBO and ENSO we will expand the discussion supported by references. We are expanding the GW discussion as well, supporting it with references since it is a study that the authors intend to study next. We will share this part of the discussion in the following reply.*

I noticed a mismatch in the presented data between Fig. 1d that shows meridional wind data above 70 km only (where the wind is southward) and Fig. 4c and 4d where, I think, this same data was partitioned regarding Ap and that shows meridional wind data also below 70 km altitude, and the wind is suddenly northward.

*It is correct that there is a difference in the data shown, mainly because for the first part (wind climatologies) we only used SMR for the meridional component as it is well covered by SMRs, but for the geomagnetic activity study, we wanted to extend the altitude range and also implemented the data from the PRRs. Part of this difference is because of the objective of the study and the difference between instruments. While the zonal wind is strong and shows strong agreement, the meridional wind is weaker and the comparison is not that good. Wilhelm et al. (2017) studied these differences. We will add more information in the data section explaining this in more detail.*

I also wonder why a larger Ap threshold of 20 is used for mid latitudes than for high latitudes. The authors refer to different geomagnetic latitudes of the radar sites, but I don't see why this is an argument. I would appreciate more substantial explanations and discussions, e.g. of the mechanisms of how wind maxima

are a "proxy for MLT dynamics", or how geomagnetic activity affects wind maxima. In Fig. 4b, is the enhancement in 2020 related to any major solar event? Do other studies exist as it seems to be a major effect?

*The different thresholds are for the following reasons. The idea was to follow Jacobi et al. (2021) who used Ap 20 for middle latitudes and we extend the analysis to high latitudes. For middle latitudes, this value reliably discriminates strong geomagnetic distortions from quiet periods. For high latitudes, it was found by Renkwitz et al. (2017), that the majority of the particle precipitation events already occur at kp=3 (~ Ap=15). Larger distortions will affect more southern locations. Given the fairly low solar activity in the last cycles and to get a sufficient amount of data for robust statistics we tend to use Ap=15 instead of Ap=20, which is used for the longer time series at middle latitudes (Fig. 6,7). We'll modify lines 132-135 to make this more clear.*

*Regarding the year 2020, it looks enhanced due to the low amount of days with geomagnetic activity above the used limit (see Fig. 7). We could remove the data from the figure, but Figure 6 intent is to have a visual behavior of the winds under low and high geomagnetic activity. We will add this comment to the text.*

Regarding the spectral analysis I wonder why a generalized Lomb-Scargle analysis is preferred over a Fourier transform. Isn't the data evenly spaced? The authors do not mention measurement uncertainties. I guess they are smaller than the variability. In Fig. 3, significance levels could have been added to the plots.

*Indeed the data is evenly spaced, but for the time series from the PRR Juliusruh, we have a missing year (2000). To avoid interpolation over the missing value and inconsistency, we made use of Lomb-Scargle analysis. We also compared to Fourier transform and checked the values were in agreement with the Fourier transform for all the time series. Mossad et al. (2023) compared both methods and found that LS is slightly more accurate for estimating the amplitude of a single frequency in the presence of minor gaps. The only disadvantage of LS compared to FFT in our case is computation time which is not really an issue.*

*The monthly variability is used as uncertainty since it is bigger than the instrument's uncertainties, as the reviewer correctly assumed. We mention this on lines 120-125, and due to the FAP being only related to the main peak, we did not want to add it to the figure to avoid common confusion with the rest of the peaks. We will add the comment of the uncertainties being smaller than the monthly variability.*

I come back to the author's nomenclature of "the maximum velocity amplitude" as their proxy for MLT dynamics which confused me. I expected an "amplitude" to relate to an oscillation, for example a tide or a gravity wave. The meaning of "maximum" was unclear, it could have related to a period of time, or altitude, or a peak-to-peak amplitude.. ? In l. 79 the authors write "the maximum velocity amplitude of the horizontal winds, independent of altitude, variability, and trends..", and I wondered how this value could be independent of altitude, variability and trends. Then, in l. 111, "the maximum amplitude of the velocity per month" seemed to indicate some deviation from a monthly mean. And indeed in l. 115 a reference to "monthly median values" was made, but Fig. 2 shows maximum values of wind components, and not amplitudes (that is differences of wind values) of any kind. I suggest to improve the language in these descriptions or add more details, and not use the term "amplitude".

*Thank you for this comment. Indeed we will change it to make it clearer. In the case of the velocity maxima being independent of altitude, we refer to the fact that we did not use fixed altitudes to obtain the wind value. Regarding the "amplitude", we already answered and explained above.*

A similar language problem might apply to "zonal mean wind" (l. 196 and others), which implies a global zonal mean, when in fact the authors probably meant "mean zonal wind". Also, the terms "a mid latitude" or "a high latitude" might be more honest than generally speaking of "mid latitudes" or "high latitudes", as only data from sites at one specific longitude was used.

*It will be corrected.*

In general, grammar could be improved. Examples are "Below the mesosphere is located the stratosphere..", "linear functions were adjusted to…" instead of "fitted to", "periodograms were extracted…" instead of "calculated", "The zonal component is built with the combination of two datasets ". Some sentences lack subjects, e.g. in l. 91, l. 138, l. 308 or in Fig. 6 caption. Specific comments and questions are listed by line number:

line 4: "mainly focuses on the summer season" what part of the study does not focus on the summer season?

*Thanks for highlighting this. The climatologies are covering the full year, but we understand the point and we will change it.*

line 6: is there no northward wind? Northward wind is mentioned in l. 16

*Indeed there is a northward wind, e.g. during spring and autumn, but for the first part of the study focusing on the summer wind maxima, the northern component does not have a distinct maximum. Nonetheless, we still investigated the meridional wind northern component for a potential geomagnetic influence.*

line 23: 1980s

*Corrected, thanks.*

line 28: regarding the broad subject of greenhouse gas monitoring, the authors only cite the works of two authors from the author's institute

*Thanks for pointing this out, we will add more references.*

line 91: FMCW is not defined. In l. 103, abbreviations are defined that are not used a second time

*FMCW stands for Frequency Modulated Continues Wave, a technique that was used for the former Juliusruh PPR. System descriptions and further references can be found in Singer et al. (1992). Since the modernization in 2001 a pulsed PRR is used.*

line 104: what is the horizontal dimension of the observational volume?

*Monostatic SMRs typically cover about 250km radius, while for the combination to radar networks it depends on the number of systems and their separation reaching 500km and above.*

line 111: do the "different ranges in the … data used for the climatologies" refer to the range of the color bar in the plot?

*It refers to the altitude range, using different instruments to capture the wind maxima where they have the best capabilities. We will make it more clear. Thanks.*

line 130: what is meant by the "complete" 19-year time series?

*It just means the full time series. We will remove the word. Thanks.*

line 134: what was the result of this study? How did the MLT respond to the change in the index?

*We answer this above, when we explained the different threshold of the Ap taken for different latitudes.*

line 140: what is meant by "the complete time series from the selected Ap index"? Isn't it rather data from all days with Ap index values above or below the respective threshold?

*Indeed, thanks, we will make it clear.*

line 154: "at the mesopause"

*Thanks.*

line 230: the study contradicting the author's results, was it done at a different longitude?

*No. One of the radars used for this study is the same that we are combining to obtain our high latitudes time series of the eastward and southward wind maxima. Even that the trend does not agree with ours, in Figure 2d between 2004 and 2012 a trend is visible that agrees with their study. This is explained in lines 231-234.*

line 262: what is a "missing solar cycle"? Do you mean conditions of solar minimum?

The missing solar cycle effect comes from Hervig et al. (2019). We will rephrase to be more clear. Thanks.

line 295: please add year range

*We will add it, thanks.*

line 300: "three studied months" this could me misinterpreted in two ways. First, the dataset is larger than three months, and second, the decline does not occur over the course of three months.

*We understand, and we will rephrase. Thanks.*

*Citations:*

*Dawkins, E. C. M., G. Stober, D. Janches, J. D. Carrillo-Sánchez, R. S. Lieberman, Ch. Jacobi, T. Moffat-Griffin, N. J. Mitchell, N. Cobbett, P. P. Batista, V. F. Andrioli, R. A. Buriti, D. J. Murphy, J. Kero, N. Gulbrandsen, M. Tsutsumi, A. Kozlovsky, J. H. Kim, C. Lee, and M. Lester, 2023: Solar cycle and long-term trends in the observed peak of the meteor altitude distributions by meteor radars, Geophys. Res. Lett., 50, e2022GL101953. https://doi.org/10.1029/2022GL101953.*

*Jacobi, C., Lilienthal, F., Korotyshkin, D., Merzlyakov, E., and Stober, G.: Influence of geomagnetic disturbances on mean winds and tides in the mesosphere/lower thermosphere at midlatitudes, Adv. Radio Sci, 19, 185–193, https://doi.org/10.5194/ars-19-185-2021, 2021.*

*Wilhelm, S., Stober, G., and Chau, J. L.: A comparison of 11-year mesospheric and lower thermospheric winds determined by meteor and MF radar at 69 ° N, Ann. Geophys., 35, 893–906, https://doi.org/10.5194/angeo-35-893-2017, 2017.*

*Renkwitz, T. and Latteck, R.: Variability of virtual layered phenomena in the mesosphere observed with medium frequency radars at 69°N, Journal of Atmospheric and Solar-Terrestrial Physics, 163, 38–45, https://doi.org/10.1016/j.jastp.2017.05.009, 2017.*

*Peters, D. H., Entzian, G., and Keckhut, P.: Mesospheric temperature trends derived from standard phase-height measurements, Journal of Atmospheric and Solar-Terrestrial Physics, 163, 23–30, https://doi.org/10.1016/j.jastp.2017.04.007, 2017.*

*Hervig, M. E., Siskind, D. E., Bailey, S. M., Merkel, A. W., DeLand, M. T., & Russell, J. M. (2019). The missing solar cycle response of the polar summer mesosphere. Geophysical Research Letters, 46, 10132–10139. https://doi.org/10.1029/2019GL083485.*

*W. Singer, P. Hoffmann, D. Keuer, R. Schminder und D. Kürschner, Wind in the middle atmosphere with partial reflection measurements during winter and spring in middle Europe, Adv. Space Res., 12(10), 299-302, 1992.*

*Mossad, M., Strelnikova, I., Wing, R., and Baumgarten, G.: Assessing Atmospheric Gravity Wave Spectra in the Presence of Observational Gaps, EGUsphere [preprint], https://doi.org/10.5194/egusphere-2023-1598, 2023.*

---

## Author Response (AR1)

**Response to RC1**: ['Comment on egusphere-2023-1465'](), Anonymous Referee #1, 08 Aug 2023.

*We appreciate the valuable comments given by the reviewer. We repeat the reviewer's concerns and provide our respective responses in italics.*

In this work, the authors analyze radar wind datasets from two sites at different latitudes, infer summer wind maxima in wind components and derive trends. They find a robust strengthening of the mesospheric westward wind at mid latitudes. The influence of daily varying geomagnetic activity on the winds is studied and excluded as source for significant trends. Gravity waves are suggested as cause of the trends.

The paper is well structured and the figures are clear. In few places there are descriptions that can be confusing, mainly when the authors speak of "maximum velocity amplitudes" when actually absolute wind values are meant, which is likely a language problem. I notice that title is a bit unspecific, being almost identical to the special issue title.

*We understand the confusion with the nomenclature. When we observe a monthly median vertical profile of the horizontal velocities, the wind profiles look like a wave, where the maximum velocity coincides with the amplitude of such "wave". From there we started to call it amplitude. We removed the 'amplitude' word in the manuscript.*

*The title has been changed to "Long-term studies of the summer wind in the mesosphere and lower thermosphere at middle and high latitudes over Europe".*

The authors list a number of similar observational studies, sometimes based on the same datasets, that sometimes come to contradicting results regarding the trends. This is attributed to different years or altitude ranges. If trends depend so sensitively on the selection of years or altitude ranges, this should be investigated in more detail to be of scientific value. For example, if altitude of wind maxima change with time, this could have been documented.

*Indeed there are studies indicating that the MLT heights have decreased in the past decades (Peters et al. 2017, Dawkins et al. 2023). Having this knowledge in mind, we used the wind maxima and not the altitude as the object of study. We did explore the altitudes of the wind maxima and found no trend in the 19-year time series. For the first years of the longer time series (33 years), the accuracy of measured altitudes of this initial radar system is not as good as in the more recent data. So in the case of testing for a trend, we only consider the years after 2004. We haven't included the plots since there are no significant trends, but indeed they show variability over the years. Below are the corresponding Figures, in red colors are the*

[Figure]

*eastward wind maxima, in purple are the westward wind maxima and in blue are the southward wind maxima. The altitude uncertainty is not included, but their values depend on the instruments as follows: Middle latitudes eastward and southward +- 1km, westward +- 5km. High latitudes eastward and southward +- 2km westward 2km. We have included the comment in lines 201-202.*

Another possible source of variability, longitude, is not mentioned at all. The authors generally refer to "middle and high latitudes" and "zonal mean winds" and therefore make the impression that their results are valid for all longitudes. This however cannot be inferred from the datasets the authors used, and some discussion of this topic should be included. Whether the obtained results are only valid locally or in the zonal mean is relevant.

*Thank you for highlighting this. Indeed our results are local since the radars are in a fixed location. We did mention this in section 2.1 lines 106-107 (preprint). However, it is known that the zonal climatologies are a good representation of global behavior, particularly at middle and high latitudes in the summer (e.g., Jacobi et al 2001). We have included this comment in the discussion, lines 269-274.*

No references to model studies were made, and models were mentioned only briefly in one sentence at the end of the manuscript. The work is mainly a report of measurements without detailed exploration of the findings regarding the underlying mechanisms. For example, periodograms with periods of 2-4 and 11 years are presented but discussion with references to QBO and ENSO remain very vague. The same applies to gravity waves being the suggested cause of the trends, but no proposals or attempts were made at how this could be tested or verified. If this is out of scope, but similar studies are available, that sometimes agree, sometimes contradict, the value of presenting measurements only is limited.

*We have included more references to works using models in the discussion briefly (lines 249-252). Sadly there are not many studies with models that represent the MLT summer well enough in order to compare it directly. In the case of QBO and ENSO we will expand the discussion supported by references (lines 278-283, 286-292). We have expanded the GW discussion as well, supporting it with references since it is a study that the authors pursue in the near future (lines 338-346).*

I noticed a mismatch in the presented data between Fig. 1d that shows meridional wind data above 70 km only (where the wind is southward) and Fig. 4c and 4d where, I think, this same data was partitioned regarding Ap and that shows meridional wind data also below 70 km altitude, and the wind is suddenly northward.

*It is correct that there is a difference in the data shown, mainly because for the first part (wind climatologies) we only used SMR for the meridional component as it is well covered by SMRs, but for the geomagnetic activity study, we wanted to extend the altitude range and also implemented the data from the PRRs. Part of this difference is because of the objective of the study and the difference between instruments. Mainly the SMR winds represent a larger volume than the PRR winds. The mean zonal wind is not affected much due to this difference in volume. However, the meridional wind presents a relatively larger latitudinal dependence and therefore the volume difference is more important. While the zonal wind is strong and shows strong agreement, the meridional wind is weaker and the comparison is not that good. Wilhelm et al. (2017) studied these differences. We will add more information in the data section explaining this in more detail. Lines 237-239 and 109-116.*

I also wonder why a larger Ap threshold of 20 is used for mid latitudes than for high latitudes. The authors refer to different geomagnetic latitudes of the radar sites, but I don't see why this is an argument. I would appreciate more substantial explanations and discussions, e.g. of the mechanisms of how wind maxima are a "proxy for MLT dynamics", or how geomagnetic activity affects wind maxima. In Fig. 4b, is the enhancement in 2020 related to any major solar event? Do other studies exist as it seems to be a major effect?

*The different thresholds are for the following reasons. The idea was to follow Jacobi et al. (2021) who used Ap≥20 for middle latitudes and we extend the analysis to high latitudes. For middle latitudes, this value reliably discriminates strong geomagnetic distortions from quiet periods. For high latitudes, it was found by Renkwitz et al. (2017), that the majority of the particle precipitation events already occur at kp=3 (~ Ap=15). Larger distortions will affect more southern locations. Given the fairly low solar activity in the last cycles and to get a sufficient amount of data for robust statistics we tend to use Ap≥15 instead of Ap≥20, which is used for the longer time series at middle latitudes (Fig. 6,7). We have added more explanation in lines 148-149 to make this difference understandable.*

*Regarding the year 2020, it looks enhanced due to the low amount of days with geomagnetic activity above the used limit (see Fig. 7). We could remove the data from the figure, but Figure 6 intent is to have a visual behavior of the winds under low and high geomagnetic activity. We added this comment in lines 217-219.*

Regarding the spectral analysis I wonder why a generalized Lomb-Scargle analysis is preferred over a Fourier transform. Isn't the data evenly spaced? The authors do not mention measurement uncertainties. I guess they are smaller than the variability. In Fig. 3, significance levels could have been added to the plots.

*Indeed the data is evenly spaced, but for the time series from the PRR Juliusruh, we have a missing year (2000). To avoid interpolation over the missing value and inconsistency, we made use of Lomb-Scargle analysis. We also compared to Fourier transform and checked the values were in agreement with the Fourier transform for all the time series. Mossad et al. (2023) compared both methods and found that LS is slightly more accurate for estimating the amplitude of a single frequency in the presence of minor gaps. The only disadvantage of LS compared to FFT in our case is computation time which is not really an issue. We added the comment in lines 135-139.*

*The monthly variability is used as uncertainty since it is bigger than the instrument's uncertainties, as the reviewer correctly assumed. We mention this on lines 120-125 (preprint version, revised version 129-135), and due to the False Alarm Probability being only related to the main peak, we did not want to add it to the figure to avoid confusion with the rest of the peaks. We added the comment of the uncertainties being smaller than the monthly variability in lines 130-131.*

I come back to the author's nomenclature of "the maximum velocity amplitude" as their proxy for MLT dynamics which confused me. I expected an "amplitude" to relate to an oscillation, for example a tide or a gravity wave. The meaning of "maximum" was unclear, it could have related to a period of time, or altitude, or a peak-to-peak amplitude.. ? In l. 79 the authors write "the maximum velocity amplitude of the horizontal winds, independent of altitude, variability, and trends..", and I wondered how this value could be independent of altitude, variability and trends. Then, in l. 111, "the maximum amplitude of the velocity per month" seemed to indicate some deviation from a monthly mean. And indeed in l. 115 a reference to "monthly median values" was made, but Fig. 2 shows maximum values of wind components,

and not amplitudes (that is differences of wind values) of any kind. I suggest to improve the language in these descriptions or add more details, and not use the term "amplitude".

*Thank you for this comment. We changed it to make it clearer. In the case of the velocity maxima being independent of altitude, we refer to the fact that we did not use fixed altitudes to obtain the wind value (lines 81-82). Regarding the "amplitude", we already answered and explained above.*

A similar language problem might apply to "zonal mean wind" (l. 196 and others), which implies a global zonal mean, when in fact the authors probably meant "mean zonal wind". Also, the terms "a mid latitude" or "a high latitude" might be more honest than generally speaking of "mid latitudes" or "high latitudes", as only data from sites at one specific longitude was used.

*It was be corrected. We refer as "winds" when the correct name is mean winds, to avoid distracting the reader with mean and median depending on what we are referring to. This is explained in lines 109-113.*

In general, grammar could be improved. Examples are "Below the mesosphere is located the stratosphere..", "linear functions were adjusted to…" instead of "fitted to", "periodograms were extracted…" instead of "calculated", "The zonal component is built with the combination of two datasets ". Some sentences lack subjects, e.g. in l. 91, l. 138, l. 308 or in Fig. 6 caption.

*We have revised the mentioned sentences and changed them.*

Specific comments and questions are listed by line number:

line 4: "mainly focuses on the summer season" what part of the study does not focus on the summer season?

*Thanks for highlighting this. The climatologies are covering the full year, but we understand the point and we changed it. (L. 4)*

line 6: is there no northward wind? Northward wind is mentioned in l. 16

*Indeed there is a northward wind, e.g. during spring and autumn, but for the first part of the study focusing on the summer wind maxima, the northern component does not have a distinct maximum. Nonetheless, we still investigated the meridional wind northern component for a potential geomagnetic influence.*

line 23: 1980s

*Corrected, thanks. (L 24)*

line 28: regarding the broad subject of greenhouse gas monitoring, the authors only cite the works of two authors from the author's institute

*Thanks for pointing this out, we will added more references. (L 29-30)*

line 91: FMCW is not defined. In l. 103, abbreviations are defined that are not used a second time

*FMCW stands for Frequency Modulated Continues Wave, a technique that was used for the former Juliusruh PPR. System descriptions and further references can be found in Singer et al. (1992). Since the modernization in 2001 a pulsed PRR is used. We have added the full name. (L 95)*

line 104: what is the horizontal dimension of the observational volume?

*Monostatic SMRs typically cover about 250km radius, while for the combination to radar networks it depends on the number of systems and their separation reaching 500km and above. We added the comment in lines 109-110.*

line 111: do the "different ranges in the … data used for the climatologies" refer to the range of the color bar in the plot?

*It refers to the altitude range, using different instruments to capture the wind maxima where they have the best capabilities. We made it more clear. Thanks. (L 121)*

line 130: what is meant by the "complete" 19-year time series?

*It just means the full time series. We removed the word. Thanks. (L 144)*

line 134: what was the result of this study? How did the MLT respond to the change in the index?

*We answer this above, when we explained the different threshold of the Ap taken for different latitudes. (L 148-149)*

line 140: what is meant by "the complete time series from the selected Ap index"? Isn't it rather data from all days with Ap index values above or below the respective threshold?

*Indeed, thanks, we removed the word "complete". (L 154)*

line 154: "at the mesopause"

*Thanks.(L 168)*

line 230: the study contradicting the author's results, was it done at a different longitude?

*No. One of the radars used for this study is the same that we are combining to obtain our high latitudes time series of the eastward and southward wind maxima. Even that the trend does not agree with ours, in Figure 2d between 2004 and 2012 a trend is visible that agrees with their study. This is explained in lines 231-234 (preprint, 258-260 revised version).*

line 262: what is a "missing solar cycle"? Do you mean conditions of solar minimum?

*The missing solar cycle effect comes from Hervig et al. (2019). We have rephrased it to be more clear. Thanks. (L 296-298)*

line 295: please add year range

*We have added it, thanks. (L 354, 361, 364)*

line 300: "three studied months" this could me misinterpreted in two ways. First, the dataset is larger than three months, and second, the decline does not occur over the course of three months.

*We understand, and we have changed it. Thanks.(L 360-362)*

*Citations:*

*Dawkins, E. C. M., G. Stober, D. Janches, J. D. Carrillo-Sánchez, R. S. Lieberman, Ch. Jacobi, T. Moffat-Griffin, N. J. Mitchell, N. Cobbett, P. P. Batista, V. F. Andrioli, R. A. Buriti, D. J. Murphy, J. Kero, N. Gulbrandsen, M. Tsutsumi, A. Kozlovsky, J. H. Kim, C. Lee, and M. Lester, 2023: Solar cycle and long-term trends in the observed peak of the meteor altitude distributions by meteor radars, Geophys. Res. Lett., 50, e2022GL101953. https://doi.org/10.1029/2022GL101953.*

*Jacobi, C., Lilienthal, F., Korotyshkin, D., Merzlyakov, E., and Stober, G.: Influence of geomagnetic disturbances on mean winds and tides in the mesosphere/lower thermosphere at midlatitudes, Adv. Radio Sci, 19, 185–193, https://doi.org/10.5194/ars-19-185-2021, 2021.*

*Wilhelm, S., Stober, G., and Chau, J. L.: A comparison of 11-year mesospheric and lower thermospheric winds determined by meteor and MF radar at 69 ° N, Ann. Geophys., 35, 893–906, https://doi.org/10.5194/angeo-35-893-2017, 2017.*

*Renkwitz, T. and Latteck, R.: Variability of virtual layered phenomena in the mesosphere observed with medium frequency radars at 69°N, Journal of Atmospheric and Solar-Terrestrial Physics, 163, 38–45, https://doi.org/10.1016/j.jastp.2017.05.009, 2017.*

*Peters, D. H., Entzian, G., and Keckhut, P.: Mesospheric temperature trends derived from standard phase-height measurements, Journal of Atmospheric and Solar-Terrestrial Physics, 163, 23–30, https://doi.org/10.1016/j.jastp.2017.04.007, 2017.*

*Hervig, M. E., Siskind, D. E., Bailey, S. M., Merkel, A. W., DeLand, M. T., & Russell, J. M. (2019). The missing solar cycle response of the polar summer mesosphere. Geophysical Research Letters, 46, 10132–10139. https://doi.org/10.1029/2019GL083485.*

*W. Singer, P. Hoffmann, D. Keuer, R. Schminder und D. Kürschner, Wind in the middle atmosphere with partial reflection measurements during winter and spring in middle Europe, Adv. Space Res., 12(10), 299-302, 1992.*

*Mossad, M., Strelnikova, I., Wing, R., and Baumgarten, G.: Assessing Atmospheric Gravity Wave Spectra in the Presence of Observational Gaps, EGUsphere [preprint], https://doi.org/10.5194/egusphere-2023-1598, 2023.*

*Jacobi, C., Lange, M., Kürschner, D., Manson, A., and Meek, C.: A long-term comparison of saskatoon MF radar and collm LF D1 mesosphere-lower thermosphere wind measurements, Physics and Chemistry of the Earth, Part C: Solar, Terrestrial & Planetary Science, 26, 419–424, https://doi.org/https://doi.org/10.1016/S1464-1917(01)00023-X, 2001.*

**Response to RC2**: 'Comment on egusphere-2023-1465', Anonymous Referee #2, 21 Aug 2023.

*We appreciate the comments and references given by the reviewer. We repeat the reviewer's concerns and provide our respective responses in italics.*

This study is focusing on the long-term trend of the horizontal wind in the MLT, using multiple radar datasets in high and middle latitudes. The calculation of the trends of horizontal wind is always challenging due to the large variability of the winds. These results will contribute greatly to the community's understanding on the MLT dynamics changes induced by increasing CO2 and climate change in the lower atmosphere. In addition, the author analyzed the radar date during geomagnetic quiet time and active time and revealed the impacts of the geomagnetic activity on the horizontal wind. This is an important topic that has not been studied very much in the community. There are some recent first principle model investigations, using TIME-GCM, on the MLT wind responses to the geomagnetic storms, showing dramatic variations in the MLT wind field during geomagnetic activities (Li et al., 2019, 2023). They could be helpful for the understanding of these results and the discussion section in this paper.

*Thanks for the references, we have included them in the discussion of the paper, lines 310-322.*

One the other hand, the algorithm is quite different to the traditional multi-linear regress approach. I encourage the author to conduct a separate multi-linear regression analysis to look at specifically the geomagnetic effects, and compare them with the current results. Because the wind results of high geomagnetic activity are based on considerably less number of days than those of low geomagnetic activity. It would be very interesting to see if they are consistent with each other.

*Indeed it's a different approach, which was our objective. The recommendation is valuable and we would like to explore it in future work, using a more traditional multiple linear regression approach including other regressors such as the geomagnetic activity.*

In addition, since there are two types of radar involved in this study (MF radar and MWR for zonal wind data), the author should be aware of the potential bias between the two instruments (MF radar underestimates the winds) and address the possible effects on the results presented. Reid et al (2018) has some insightful discussion on this topic, and the paper could be beneficial for the author to clarify this issue.

*Indeed, we are aware of this difference. The PRR winds for high latitudes are corrected based on the Angle-of-Arrival statistics and compared to mesospheric VHF wind measurements (Renkwitz et al., 2018). However, we are not using data above 80km with PRR, where these differences could be more significant. We have added the comment in section 2.1 (lines 109-116). Thanks.*

*T. Renkwitz, M. Tsutsumi, F. I. Laskar, J. L. Chau und R. Latteck, On the role of anisotropic MF/HF scattering in mesospheric wind estimation, Earth Plan. Space, 70:158, doi:10.1186/s40623-018-0927-0, 2018.*

There is not much investigations on the inter-annual oscillations and solar cycle effects in the current manuscript. I suggest removing 4.1 and 4.2, just focusing on wind trends and geomagnetic( effects.

*We agree that it needs more discussion. We find these results interesting since the amplitudes of the oscillations are highly variable over the years and vary with altitude. We have added more discussion with references (lines 278-283, 286-290).*

Minor issues:

The author keeps using the term "velocity amplitude" throughout the paper. This is confusing, since "amplitude" is usually referring to the magnitude of the modulation, such as wave amplitude, but this study is about the mean wind. I think this is due to some writing habit. Please revise.

*We have removed the word amplitude and kept 'maximum wind velocity'.*

Page 1, line 4: "…absence of intense planetary wave…". This is incorrect. The QTDW is quite active in the summer hemisphere. But it should not affect the results, since the author is using the sliding 16-day window.

*Indeed, although the QTDW starts in July and is not active during the entire summer. As the reviewer mentioned any effect is removed with the 16-day running window. We have changed, thanks. (L 4)*

Page 3, line 78, The MLT height decrease is also revealed by Yuan et al., 2019 on the trend of the mesopause height, a direct evidence of "shrinking" of the upper atmosphere.

*We have added the reference. (L 80)*

Page 5, the definition of low and high geomagnetic quiet days seems arbitrary. For high latitude AP >= 15, but for midlatitude AP>=20. I understand that you need stronger geomagnetic activity to see the changes at midlatitudes, but I think the criteria should be consistent. In addition, it is expected to see weaker responses at midlatitudes than at high latitudes.

*The AP limit was taken from Jacobi et al. (2021) for middle latitudes. We chose Ap >= 15 following the paper by Renkwitz and Latteck (2017) for the high latitudes. The difference between 15 and 20 at high latitudes, is not significant, but as mentioned by the reviewer, the amount of days used in the statistics decreases for AP>=20. Considering the shorter time series at high latitudes, we found this change better, but we wanted to continue the values for middle latitudes, as a follow-up and extension of the work done by Jacobi et al. (2021). We added more explanation, lines 148-149.*

Page 9, line 198-199, this is expected from the model simulations mentioned above. But the high latitude responses are more complex than those at the midlatitudes due to aurora heating.

*We have added the suggested references. (L 310-322)*

Page 13, why leaves a gap in each of figure 6a and 6b?

*This is to keep the same height format as the high latitude height range (Figures 5a and 5b).*

Page 14, I suggest deleting the statement "In addition,…", because there is not much investigation on this topic, just some hypothesis. See my comment above.

*We have changed the sentence. (L 350-351)*

Page 15, "… the contribution of planetary waves is ….." see my comment above.

*We have changed it. (L 336-338)*

Suggested references:

Li, J., Wei, G., Wang, W., Luo, Q., Lu, J., Tian, Y., Xiong, S., Sun, M., Shen, F., Yuan, T., Zhang, X., Fu, S., Li, Z., Zhang, H., Yang, C. (2023). A modeling study on the responses of the mesosphere and lower thermosphere (MLT) temperature to the initial and main phases of geomagnetic storms at high latitudes. Journal of Geophysical Research: Atmospheres, 128, e2022JD038348. https://doi.org/10.1029/2022JD038348

Li, J., W. Wang, J. Lu, J. Yue, A. Burns, T. Yuan, X. Chen and W. Dong (2019): A modeling study of the responses of mesosphere and lower thermosphere (MLT) winds to geomagnetic storms at middle latitudes, J. Geophys. Res. Space Physics 124. https://doi.org/10.1029/2019JA026533.

Reid, I.M., McIntosh, D.L., Murphy, D.J. et al. Mesospheric radar wind comparisons at high and middle southern latitudes. Earth Planets Space 70, 84 (2018). https://doi.org/10.1186/s40623- 018-0861-1

Yuan, T., Solomon, S. C., She, C.-Y., Krueger, D. A., & Liu, H.-L. ( 2019). The long-term trends of nocturnal mesopause temperature and altitude revealed by Na lidar observations between 1990 and 2018 at mid-latitude. Journal of Geophysical Research: Atmospheres, 124. https://doi.org/10.1029/2018JD029828.